# Super-resolving microscopy reveals the localizations and movement dynamics of stressosome proteins in *Listeria monocytogenes*

Buu Minh Tran [1], Dmitrii Sergeevich Linnik[1], Christiaan Michiel Punter[1], Wojciech Mikołaj Śmigiel [1], Luca Mantovanelli[1], Aditya Iyer [1], Conor O'Byrne [2], Tjakko Abee[3], Jörgen Johansson[4] & Bert Poolman [1✉]

The human pathogen *Listeria monocytogenes* can cope with severe environmental challenges, for which the high molecular weight stressosome complex acts as the sensing hub in a complicated signal transduction pathway. Here, we show the dynamics and functional roles of the stressosome protein RsbR1 and its paralogue, the blue-light receptor RsbL, using photo-activated localization microscopy combined with single-particle tracking and single-molecule displacement mapping and supported by physiological studies. In live cells, RsbR1 is present in multiple states: in protomers with RsbS, large clusters of stressosome complexes, and in connection with the plasma membrane via Prli42. RsbL diffuses freely in the cytoplasm but forms clusters upon exposure to light. The clustering of RsbL is independent of the presence of Prli42. Our work provides a comprehensive view of the spatial organization and intracellular dynamics of the stressosome proteins in *L. monocytogenes*, which paves the way towards uncovering the stress-sensing mechanism of this signal transduction pathway.

[1] Department of Biochemistry, University of Groningen, Groningen, the Netherlands. [2] Microbiology, School of Biological & Chemical Sciences, Ryan Institute, University of Galway, Galway, Ireland. [3] Laboratory of Food Microbiology, Wageningen University & Research, Wageningen, the Netherlands. [4] Department of Molecular Biology, Umeå University, Umeå, Sweden. ✉email: b.poolman@rug.nl

The Gram-positive bacterium *L. monocytogenes* is an essential model in infection biology[1–3]. The organism is ubiquitous and thrives in diverse environments, such as in water, soil and various food products; it also can invade and grow inside human cells[4,5]. *L. monocytogenes* is not only a dangerous pathogen but is also known for its resilience to harsh conditions like high osmotic stress, low temperature, and low pH[6–9].

In *L. monocytogenes*, the alternative sigma factor B (SigB or σ[B]) plays an essential role in the general stress response and the virulence of the organism[10–15]. SigB binds to the RNA polymerase and regulates the expression of around 300 genes[16]. SigB activation is controlled by the activity of an anti-sigma factor (RsbW) and an anti-anti-sigma factor (RsbV), which creates a biochemical signalling cascade (Fig. 1). The acronym Rsb stands for Regulator of sigma B[17,18]. In this cascade, environmental stress signals are perceived upstream and mediated by a supra-macromolecular complex called the stressosome; the activity of SigB is downstream of the signal transduction cascade[16,19,20].

The stressosome plays a vital role in signal transduction in many Gram-positive[21,22] and Gram-negative bacteria[23], *Mycobacterium sp.*[24], and particularly *Bacillus subtilis*[22,25,26]. In *Bacillus subtilis*, the stressosome consists of a core scaffold comprised of the Sulfate Transporter and Anti-Sigma factor antagonist (STAS) domains of RsbS and RsbR, with the sensory domains of RsbR and its paralogues protruding from the core as 'turrets'[22]. In *L. monocytogenes*, the common RsbR (Lmo0889 or RsbR1) has four paralogues called Lmo0161, Lmo1642, Lmo1842, and Lmo0799, which are renamed as RsbR2, RsbR3 and RsbR4, and RsbL, respectively[27]. RsbL (Lmo0799) is involved in light sensing, the best-characterized function of any RsbR proteins. Given the number of different RsbR paralogues and their

dynamic integration into stressosome complexes, it is important to understand how they affect the structure, stability and dynamics of the stressosome. A single-particle cryo-EM structure of the stressosome from *L. monocytogenes* is available[21], which shows the RsbR1-RsbS-RsbT complex as an icosahedron-like architecture with the sensory domains protruding from the turrets. The stressosome formed by 20 RsbR1-dimers connects to 10 RsbS dimers and 10 RsbT dimers. The association of RsbR paralogues with the stressosome complex remains undetermined.

The available literature is contradictory on the localization of the stressosome inside bacterial cells. The current view of stressosome localization is from fixed immuno-labelled *Bacillus subtilis* cells, which shows several distinct foci consistent with around ~20 stable stressosome complexes[22]. Recently, a new miniprotein Prli42 in *L. monocytogenes* and conserved among the Firmicutes has been implicated in stressosome localization[28]. Immunoprecipitation assays showed that Prli42 is present in the plasma membrane and tethers the stressosome complex to the membrane in response to oxidative stress by interacting with RsbR1. A recent study reported that most RsbR1 and RsbL are also found on the membrane of *L. monocytogenes* in the intracellular eukaryotic environment[29]. In actively-growing cells in the brain heart infusion (BHI) medium, phosphorylated RsbR1 was found in the membrane fractions but at a low concentration[27]. However, no previous study has investigated the stressosome localization in *L. monocytogenes* live cells using microscopy, and no information is available on the mobility of the stressosome-related proteins inside living cells.

In this study, we applied photo-activated localization microscopy (PALM) to determine the localization of stressosome proteins in *L. monocytogenes* living cells. Single-particle tracking (SPT)

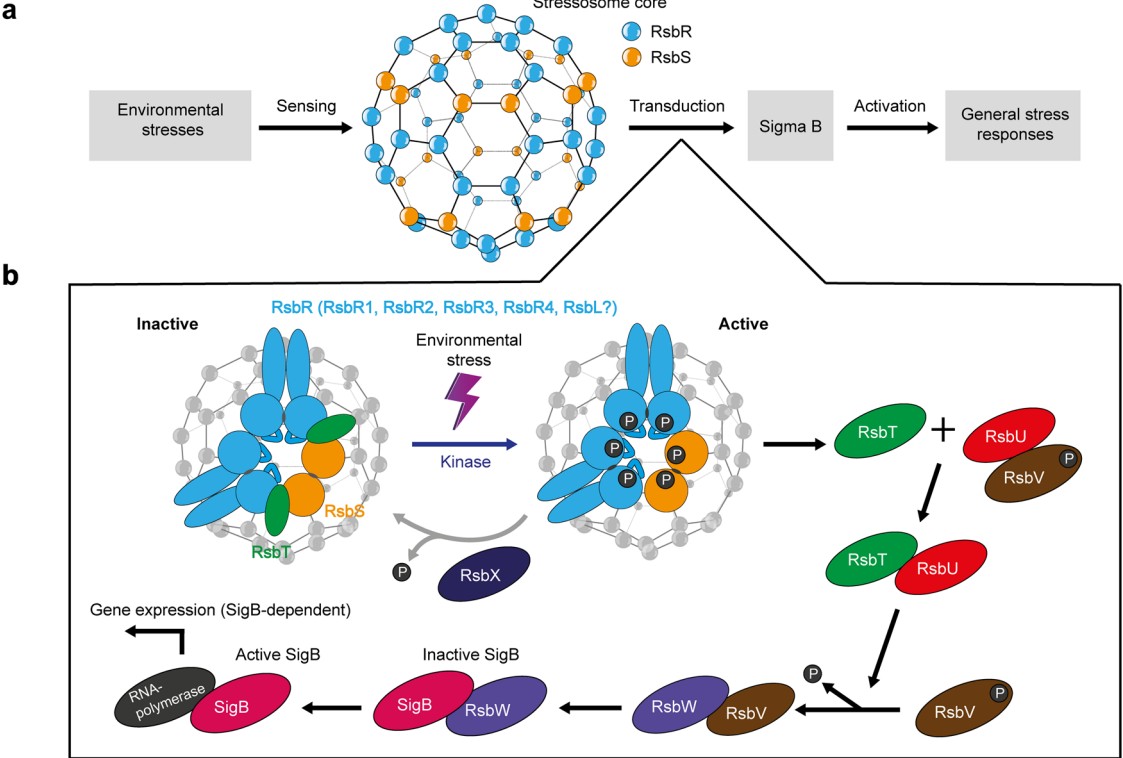

**Fig. 1 Overview of stress signal transduction in *L. monocytogenes*.** The scheme is based on data from *L. monocytogenes* and *Bacillus subtilis*. **a** Stress signal pathway from signal sensing to response via the stressosome. The stressosome core is drawn based on the cryoEM structure[21]. **b** Model of regulation of SigB activity via the stressosome. RsbT is freed from the stressosome following activation of its kinase activity upon receiving stress signals. RsbT then associates with RsbU, which triggers the dephosphorylation of RsbV and the subsequent release of SigB from RsbW. RsbX is a phosphatase that restores the sensing-ready state of the stressosome.

and single-molecule displacement mapping (SMdM) were used to measure the diffusion of the RsbR proteins. We first probed the dynamics of free cytosolic fluorescent protein (mEos3.2) and membrane-bound Prli42, then benchmarked the data against those of RsbR1 and RsbL. We find that RsbR1 diffuses slower than expected, given its molecular weight and interacts with Prli42 on the membrane or forms higher-order cytoplasmic complexes. We also show that the blue-light sensor RsbL diffuses freely in the cytoplasm but forms clusters upon exposure to light.

## Results

**Phenotypic screening of the integrative strains**. We constructed genomic integration strains for *mEos3.2::rsbR1* and *mEos3.2::rsbL* (Supplementary Fig. 1), in which the gene expression is under the native promoters to rule out possible effects of overproducing the stressosome components, and we tested their phenotype. We first compared *mEos3.2::rsbR1* with wild-type *L. monocytogenes*, *ΔrsbR1*, *ΔsigB*, and *Δprli42* by following the growth under various stress conditions, as previously described[28]. We did not find significant differences in the growth of the *Listeria* strains in the ethanol, NaCl, low pH or $H_2O_2$ stress (Supplementary Fig. 2). In the acid shock experiment[30], *ΔrsbR1* and *ΔsigB* are notably more sensitive to the acid treatment than the wild-type, integrative *mEos3.2::rsbR1*, and *Δprli42* strains (Supplementary Fig. 3). No colonies were detected after 10 min exposure to pH 2.5 (detection threshold at $10^2$ CFU.mL$^{-1}$). The wild-type, *mEos3.2::rsbR1*, and *Δprli42* displayed a similar level of acid resistance, and colonies were still detected after 30 min. Thus, the overall RsbR1 phenotypic tests suggest that the integration strain *mEos3.2::rsbR1* still has its native phenotype.

As for the blue-light sensor RsbL, we used the ring formation phenotype on low-agar under oscillating cycles of light and dark to test the integration strain *mEos3.2::rsbL*[31]. The blue light-sensing phenotype was lost when the gene encoding *mEos3.2* was added upstream of *rsbL* on the chromosome, which is indicated by a nearly homogenous opaque area (Supplementary Fig. 4). We then added *rsbL* in trans, by expressing the gene from the pNF RsbL vector. The formation of alternating opaque and translucent rings indicates the restoration of blue light sensing. Hence, the gene fusion resulting in N-terminal tagging of RsbL with mEos3.2 hampers the light sensing of the protein.

**Single-molecule displacement mapping (SMdM) and intracellular diffusion of free cytosolic mEos3.2 and membrane-bound Prli42**. We applied single-molecule displacement mapping (SMdM)[32] to obtain information on proteins and fusions at the single-molecule level by monitoring a large number of displacements (e.g. ~10,000 for free cytosolic mEos3.2) and determining heterogeneities of intracellular diffusion at the nanometer scale. Figure 2 shows the principle of this new method to probe the dynamics of proteins in the cell. Modified stroboscopic illumination (Fig. 2a) of the bacterial sample with short and high-intensity laser pulses is strictly timed so that fluorescence from particles is recorded at the end of odd and the beginning of even frames. All single-molecule localizations of mEos3.2 molecules are reconstructed in high-density point clouds (Fig. 2b), which are clustered as single cells and horizontally aligned to facilitate the map reconstruction (Fig. 2c, d). We show the SMdM data of a cell carrying mEos3.2 in Fig. 2e. The left panel displays the starting positions of detected peaks. After fitting the displacement distribution with a modified probability density function (PDF), we obtained the diffusion coefficient ($D_L$) value (i.e. 8.7 μm$^2$.s$^{-1}$ for mEso3.2) (middle panel). The diffusion map is the final result in SMdM method. In Fig. 2e (bottom right panel), we show the high-resolution map of apparent local diffusion of the mEos3.2 in

*L. monocytogenes* cell. For the details of SMdM technique, we refer to the Methods section.

Figure 3a shows representative SMdM data for membrane-bound mEos3.2-Prli42. The highly inclined and laminated optical sheet (HILO; Supplementary Fig. 5) microscopy approach enables the detection of soluble proteins in the middle of the cell, while most membrane molecules are detected at the cell periphery. The $D_L$ is 0.1 μm$^2$.s$^{-1}$ for mEos3.2-Prli42 (middle panel of Fig. 3a).

The diffusion maps show the apparent local diffusion of proteins in the cell. The free cytosolic mEos3.2 molecules diffuse relatively fast with some pixel-to-pixel variations. The cytosolic diffusion seems faster in the middle of the cell and slower in the pole regions. The $D_L$ of mEos3.2-Prli42 at the cell's periphery is 0.1 μm$^2$.s$^{-1}$. Some small regions inside the cell are blank (black pixels) because there were insufficient displacements to reconstruct diffusivity maps.

To obtain insight into possible differences in subcellular diffusion, we selected the pole regions (20% of the cell length) from the leftmost and rightmost x-coordinates. We then took the remaining (60%) as the middle region. The data fitting from these regions and the estimation of the $D_L$ are shown in Supplementary Fig. 6. We summarize the region-dependent $D_L$ values in Fig. 3b, c. Interestingly, the average $D_L$ of mEos3.2 for the entire cells is $7.5 \pm 0.6$ μm$^2$.s$^{-1}$. For the middle regions, the $D_L$ is $8.2 \pm 0.9$ μm$^2$.s$^{-1}$, and for the poles, it is $6.4 \pm 0.7$ and $5.9 \pm 0.8$ μm$^2$.s$^{-1}$. The poles thus display significantly slower diffusion than the middle regions of the cell. The average $D_L$ of mEos3.2-Prli42 for the entire cells and middle regions ($0.16 \pm 0.02$ μm$^2$.s$^{-1}$) and the poles ($0.19 \pm 0.03$ and $0.15 \pm 0.02$ μm$^2$.s$^{-1}$) are similar.

In addition to SMdM, we also used fluorescence recovery after photobleaching (FRAP) to determine the ensemble $D_L$ of the tandem fusions. The ensemble $D_L$ implies the bulk mobility of the total protein population, and it lacks single-molecule sensitivity. In the exponential phase, the ensemble $D_L$ of mEos3.2 and mEos3.2-Prli42 were $8.8 \pm 3.7$ μm$^2$.s$^{-1}$ and $0.3 \pm 0.2$ μm$^2$.s$^{-1}$ (mean ± SD), respectively (Supplementary Fig. 7a). The values in the stationary phase-grown cells were $6.3 \pm 3.3$ μm$^2$.s$^{-1}$ and $0.3 \pm 0.2$ μm$^2$.s$^{-1}$ for mEos3.2 and mEos3.2-Prli42, respectively (Supplementary Fig. 7b). The ensemble $D_L$ obtained by FRAP are in a similar range characterized by the SMdM.

These findings reinforce the idea that small cytosolic proteins like mEos3.2 diffuse unhindered by the nucleoid region and skeleton structures through the entire cytoplasm. Below we discuss the apparent slower diffusion in the pole regions. The $D_L$ of mEos3.2 and membrane-bound mEos3.2-Prli42 are used to benchmark the intracellular diffusion data of the stressosome components RsbR1 and RsbL.

We performed in-gel fluorescence to verify the constructs produced in cells (Supplementary Fig. 8). We found that mEos3.2-RsbR1 and mEos3.2-RsbL migrated faster than their predicted molecular weights ($M_W$), which are 57.8 and 55.2 kDa, respectively. The migration of these fusions was the same in wildtype and *Δprli42* strains. Surprisingly, mEos3.2-RsbR1 migrated faster than mEos3.2-RsbL despite having a slightly higher $M_W$, most likely due to incomplete unfolding in the presence of SDS[33,34] and perhaps a contribution from the binding of RsbL with the photosensitive cofactor flavin mononucleotide (FMN)[35]. The band of mEos3.2-Prli42 is at around 30 kDa, which agrees with the expected $M_W$. These results indicate that the tandem fusions inside *L. monocytogenes* are intact. Additionally, we show details of the protein sequences, theoretical molecular weight, pI value, and net charge of the fusions in Supplementary Table 1.

**RsbR1 interacts with the plasma membrane in *L. monocytogenes* via Prli42**. To investigate the intracellular diffusion and localization of RsbR1, we first constructed GFP-RsbR1 and expressed the gene

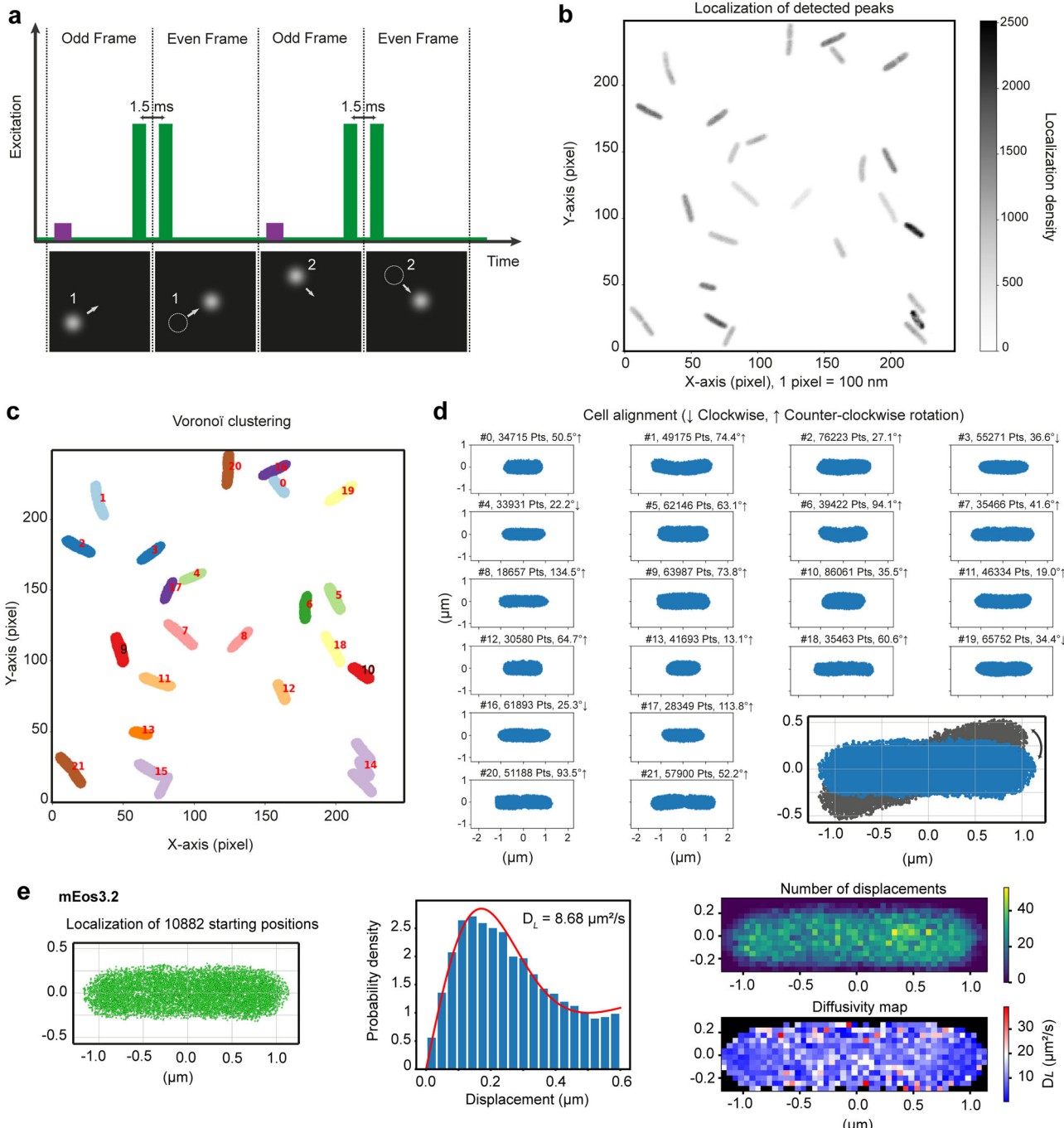

**Fig. 2 Overview of SMdM acquisition and initial steps of data analysis from *L. monocytogenes* cells carrying mEos3.2 as an example. a** Scheme of modified stroboscopic illumination for SMdM acquisition. At the beginning of odd frames, mEos3.2 is photoconverted from a green to a red state by a low-power laser pulse at 405 nm (purple bars). The excitation pulses of the 561 nm laser (green bars) peak at the end of odd frames and the beginning of even frames to create pairs of short peak-to-peak times. This illumination method is employed to track fast-moving molecules (e.g., short displacements of molecules 1 and 2 as shown in the bottom panels). **b** A 250 × 250 pixels field of view (FOV) showing all single-molecule localizations (of free mEos3.2 molecules) in the form of point clouds. **c**, The point clouds of (**b**) are clustered using the Voronoï tessellation method. Each cluster represents a cell. We ignored incorrect clusters (e.g., clusters 14 and 15 having 3 and 2 cells, respectively) due to too high density or proximity in the analysis. **d** Once clustered, the cells in (**c**) are aligned along the x-axis to facilitate map reconstruction. Each panel in (**d**) indicates the cells in (**c**) with the information of cell number, the number of points detected, and rotation degree and direction. The bottom (right panel) is an example of point cloud rotation. **e** SMdM of mEos3.2 in a cell of *L. monocytogenes*. Left panel: a point cloud of starting localizations clustered by the Voronoï method. Middle panel: distribution of all single-molecule displacements from the cell shown in the left panel. The red curve shows the fitting of the maximum likelihood estimation (MLE). Right (top) panel: spatial distribution of the number of displacements. Right (bottom) panel: a map of intracellular diffusivity with 50 × 50 nm$^2$ resolution.

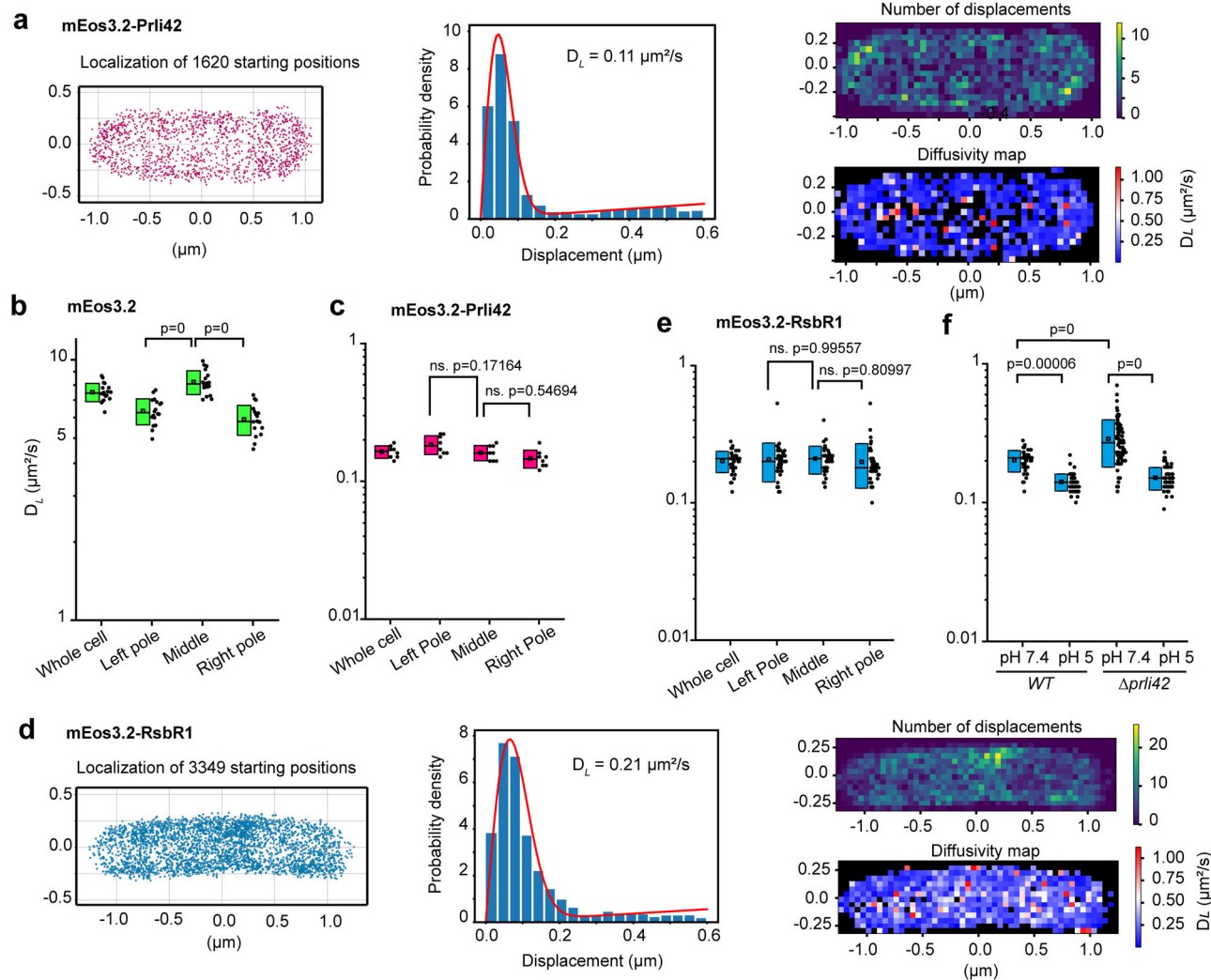

**Fig. 3 SMdM of proteins in *L. monocytogenes*. a**, and **d**, SMdM results of membrane-bound mEos3.2-Prli42 and mEos3.2-RsbR1, respectively. Left panel: a point cloud of starting localizations clustered by the Voronoï method. Middle panel: distribution of all single-molecule displacements from the cell shown in the left panel. The red curve is the maximum likelihood estimation (MLE) outcomes using Eq. (4), yielding the $D_L$ indicated in the panel. Right (top) panel: spatial distribution of the number of displacements. Right (bottom) panel: a map of intracellular diffusivity with $50 \times 50$ nm$^2$ resolution. **b**, **c**, and **e**, Box charts of the $D_L$ obtained from the cells of different replicates with free mEos3.2 ($N = 18$), mEos3.2-Prli42 ($N = 8$), and mEos3.2-RsbR1 ($N = 40$), respectively. **f**, The $D_L$ obtained from mEos3.2-RsbR1 in the wild-type and $\Delta prli42$ strains at pH 7.4 ($N = 40$ and 75, respectively) and shocked at pH 5 ($N = 59$ and 57, respectively) in the BHI medium ($D_L$ of the whole cell). Each dot shows the data of one cell. The box range indicates the standard deviation (SD), and the open square and dash symbols inside the boxes indicate the mean and median, respectively. (ns) is not significant and statistical significance was determined by one-way ANOVA, followed by Tukey's posthoc test to calculate *P*-values. The left and right poles hold 20% of the cell's length, and the remaining 60% is the middle region. Details of region selection are in Supplementary Fig. 6.

fusion from the P*dlt* promoter of the pNF vector. We observed big aggregates at one of the poles in *L. monocytogenes* EGD-e (wild-type) and the mutant strain *ΔrsbR1*. Confocal microscopy shows the aggregates as bright immobile fluorescent spots (Supplementary Fig. 9). Under native conditions, *rsbR1* is co-transcribed with *rsbS* and *rsbT* on the chromosome. We then made constructs in which *mEos3.2-rsbR1* is co-expressed with *rsbS* and *rsbT* and no longer observed fluorescent aggregates.

Figure 3d shows the SMdM data of the co-expressed mEos3.2-RsbR1 in wild-type *L. monocytogenes*. The middle panel of Fig. 3d shows that the fitting of the displacements yielded a $D_L$ of 0.21 μm$^2$.s$^{-1}$, and this value is more than an order of magnitude slower than expected for a soluble fusion protein with a $M_W$ of 57.8 kDa[36]. The poles and middle regions were selected as shown in Supplementary Fig. 6c. As observed for mEos3.2-Prli42, the $D_L$ of mEos3.2-RsbR1 in the middle ($0.21 \pm 0.05$ μm$^2$.s$^{-1}$) and

the pole regions ($0.21 \pm 0.07$ and $0.2 \pm 0.07$ μm$^2$.s$^{-1}$) are not significantly different (Fig. 3e).

RsbR1 can sense acid stress in the absence of other RsbR paralogs[30]. We tested whether the diffusion of RsbR1 is affected by acidic conditions at pH 5.0 (Fig. 3f) and observed a statistically significant difference ($P < 0.001$) in the diffusion of mEos3.2-RsbR1 in the wildtype ($D_L = 0.2 \pm 0.04$ μm$^2$.s$^{-1}$) and $\Delta prli42$ ($D_L = 0.29 \pm 0.11$ μm$^2$.s$^{-1}$) at pH 7.4. These data suggest that a fraction of RsbR1 interacts with the membrane via the mini-protein Prli42, but the differences are small and may not have much physiological importance. We also reiterate that wild-type and $\Delta prli42$ strains display a similar level of stress resistance (*vide supra;* Supplementary Fig. 2). At pH 5, RsbR1 diffuses significantly slower with $D_L = 0.14 \pm 0.02$ and $0.15 \pm 0.03$ μm$^2$.s$^{-1}$ for wildtype and $\Delta prli42$, respectively, than in cells kept at pH 7.4, which may reflect a different physical phase or viscosity of the cytoplasm at low pH stress.

Additionally, we measured the ensemble $D_L$ by using FRAP. The ensemble $D_L$ of mEos3.2-RsbR1 in exponentially growing wild-type and $\Delta prli42$ strains were at $0.75 \pm 0.62$ and $0.66 \pm 0.46$ $\mu m^2.s^{-1}$, respectively. The $D_L$ drops by 40% in the stationary phase of growth in both the wild-type and $\Delta prli42$ strain (Supplementary Fig. 7). Yet, FRAP is unable to capture the small but significant differences in diffusion observed by SMdM. We further investigated the fast and slow fractions separately by using single-particle tracking and then assigned different states of the proteins (See section **Single-particle tracking**).

**Super-resolution microscopy reveals localization of Prli42, RsbR1 and RsbL in *L. monocytogenes*.** We used PALM reconstruction to investigate the intracellular localization of Prli42, RsbR1 and RsbL. Figure 4a shows the images of mini-protein Prli42 by HILO illumination. With an acquisition of 5000 frames, the mEos3.2-Prli42 distribution is relatively homogenous around the cell, but some high-density spots are visible (white triangles). The cross-section intensity profile points to the localization of Prli42 at the cell periphery, consistent with its reported membrane association[28].

PALM images of mEos3.2-RsbR1 in the wild-type and $\Delta prli42$ (Fig. 4b, c) indicate that RsbR1 localizes at the cell periphery proximal to the membrane. However, it is striking that membrane localization is only observed in the wild-type strain and not in $\Delta prli42$ (see Supplementary Fig. 10 for additional images). The membrane localization of RsbR1 in wild-type is also more evident from the image reconstructed from long acquisitions of SMdM (Supplementary Fig. 11). We show the membrane-proximal mEos3.2-RsbR1 clusters in Supplementary Videos 1 and 2. Hence, the membrane localization of RsbR1 is Prli42 dependent. In contrast, results from PALM experiments point towards a cytoplasmic localization of mEos3.2-RsbL in the wild-type and $\Delta prli42$ (Fig. 4d, e; and Supplementary Fig. 10).

**Single-particle tracking (SPT).** SMdM and SPT provide both localization and mobility data. We categorised the protein states based on individual trajectories of particles and the obtained $D_L$; the latter gives an estimate of the size. However, it is essentially impossible to determine the exact number of diffusive states on the basis of the tracking data[37–39]. Hence, we first used the localization data (membrane or cytoplasm; different conditions) and then the SPT and SMdM (and FRAP) data to estimate the $D_L$, from which we infer whether or not the proteins cluster. We assigned different states that may correspond to different biologically relevant protein fractions in the cells. While the free (Fr) fraction indicates freely diffusing monomeric molecules, the intermediate (Int) fraction refers to the oligomeric molecules (e.g. RsbR1-RsbS protomer). The membrane-bound (mBd) fraction can be defined as molecules binding with membrane components, and clustered (Cl) fraction indicates molecules forming higher-order structures or aggregates.

In Fig. 4f, we show an example of SPT fitting the histogram of all the displacements from the trajectories of mEos3.2-Prli42 to a two-component 2D random-walk model in Eq. (6). Figure 4g shows the two components of diffusion, the membrane-bound (mBd) and clustered (Cl), for both Prli42 and RsbR1. The starred membrane-bound (mBd*) fraction refers to mEos3.2-RsbR1 molecules loosely associated with the membrane in $\Delta prli42$. The diffusion of mEos3.2-Prli42 in $\Delta prli42$ is $0.08 \pm 0.04$ and $0.02 \pm 0.01$ $\mu m^2.s^{-1}$ for the mBd and Cl fractions. In wild-type, mEos3.2-RsbR1 diffuses at $0.15 \pm 0.04$ and $0.03 \pm 0.01$ $\mu m^2.s^{-1}$ for the mBd and Cl fractions. In $\Delta prli42$, the $D_L$ of mEos3.2-RsbR1 is $0.25 \pm 0.07$ and $0.06 \pm 0.02$ $\mu m^2.s^{-1}$ for the mBd* and Cl fractions. This finding also reinforces PALM data, where the membrane localization of RsbR1 is observed as Prli42-

dependent. Figure 4h shows the percentage of the different fractions for the data shown in Fig. 4g, which are 50/50 for membrane-bound and clustered Prli42 in $\Delta prli42$, and roughly 60/40 for mBd/mBd* and Cl fractions of RsbR1 in both wild-type and $\Delta prli42$.

In both strains, the ensemble $D_L$ of mEos3.2-RsbR1 measured by FRAP (Supplementary Fig. 7) is considerably slower compared to free cytosolic proteins implying its existence in the form of protomers of RsbR1-RsbS in the living cells. We assigned this fraction as an intermediate (Int) state. Molecules of mEos3.2-RsbR1 in the Cl fraction diffuse considerably slower than those in the mBd fraction. The diffusion rate is also consistent with the bound and fully active ribosomes[39,40], suggesting its association with the stressosome complexes.

In the case of RsbL, we mark the two fractions as free (Fr) and membrane-bound (mBd). The $D_L$ values of RsbL in the wild-type are $0.90 \pm 0.12$ and $0.12 \pm 0.02$ $\mu m^2.s^{-1}$ for Fr and mBd fractions, respectively. RsbL measured in $\Delta prli42$ resulted in $D_L$ values of $0.66 \pm 0.25$ and $0.09 \pm 0.04$ $\mu m^2.s^{-1}$ for Fr and mBd fractions, respectively (Fig. 4f). The fractions of free and mBd RsbL are 50/50 and 56/44 for wild-type and $\Delta prli42$, respectively (Fig. 4g). We provide additional SPT fitting examples of RsbR1 and RsbL in Supplementary Fig. 10.

**Clustering of RsbL upon light irradiation.** In contrast to free mEos3.2, mEos3.2-Prli42, and mEos3.2-RsbR1, we observed increased clustering of mEos3.2-RsbL upon laser light irradiation (405 nm and 561 nm, violet and green) during SMdM acquisition (Fig. 5a). The clustering of RsbL is correlated with irradiation time. Additionally, the clusters localize on the cell periphery suggesting the membrane-bound fraction (Fig. 5a3 and Fig. 5a4). We reconstructed the localized molecules for 5000-frame intervals and concatenated the data into a video (Supplementary Video 3). As shown in the video, RsbL is distributed homogeneously in the cytoplasm in cells grown in the dark and the clusters started to emerge after around 4.3 min (~15,000 frames). The clusters are mobile in the cell, unlike misfolded proteins in inclusion bodies.

Due to the clustering effect, we have free molecules at the beginning of the experiment and membrane-bound clusters after irradiation. Hence, we applied two-component fitting to Eq. (4) for the SMdM data of mEos3.2-RsbL. We also wondered if the dissociation of RsbL proteins from the clusters occurs upon dark incubation for 70 min after different irradiation periods. Figure 5b shows that the free RsbL molecules diffuse an order of magnitude faster than the molecules of the bound fraction ($1.90 \pm 0.48$ $\mu m^2.s^{-1}$ *versus* $0.18 \pm 0.04$ $\mu m^2.s^{-1}$). Upon irradiation during the SMdM acquisition, the $D_L$ of the bound fraction decreased, whereas that of the free fraction increased slightly. We did not observe the dissociation of RsbL from the clusters of mBd molecules upon dark incubation (Fig. 5b, c).

We then asked whether the RsbL clustering upon laser irradiation occurs in the strains lacking Prli42. Prli42 interacts with RsbR-like proteins[28], and RsbL-like protein YtvA interacts with RsbR1-like RsbRA in *Bacillus subtilis*[41]. We performed three continuous acquisitions of 60 min with the wild-type and $\Delta prli42$ strains expressing mEos3.2-RsbL (Supplementary Fig. 13). The clustering of RsbL is not dependent on Prli42.

**SMdM of RsbL and 2-component fitting by maximum likelihood estimation (MLE).** The SMdM technique was originally developed for tracing molecules with a single diffusion component[32]. However, due to the clustering effect upon irradiation of RsbL, which leads to changes in diffusion constant, we implemented a methodology to fit multiple components (Eq. (4)). Figure 6 shows the SMdM results of mEos3.2-RsbL in a single *L. monocytogenes* cell during 150 min

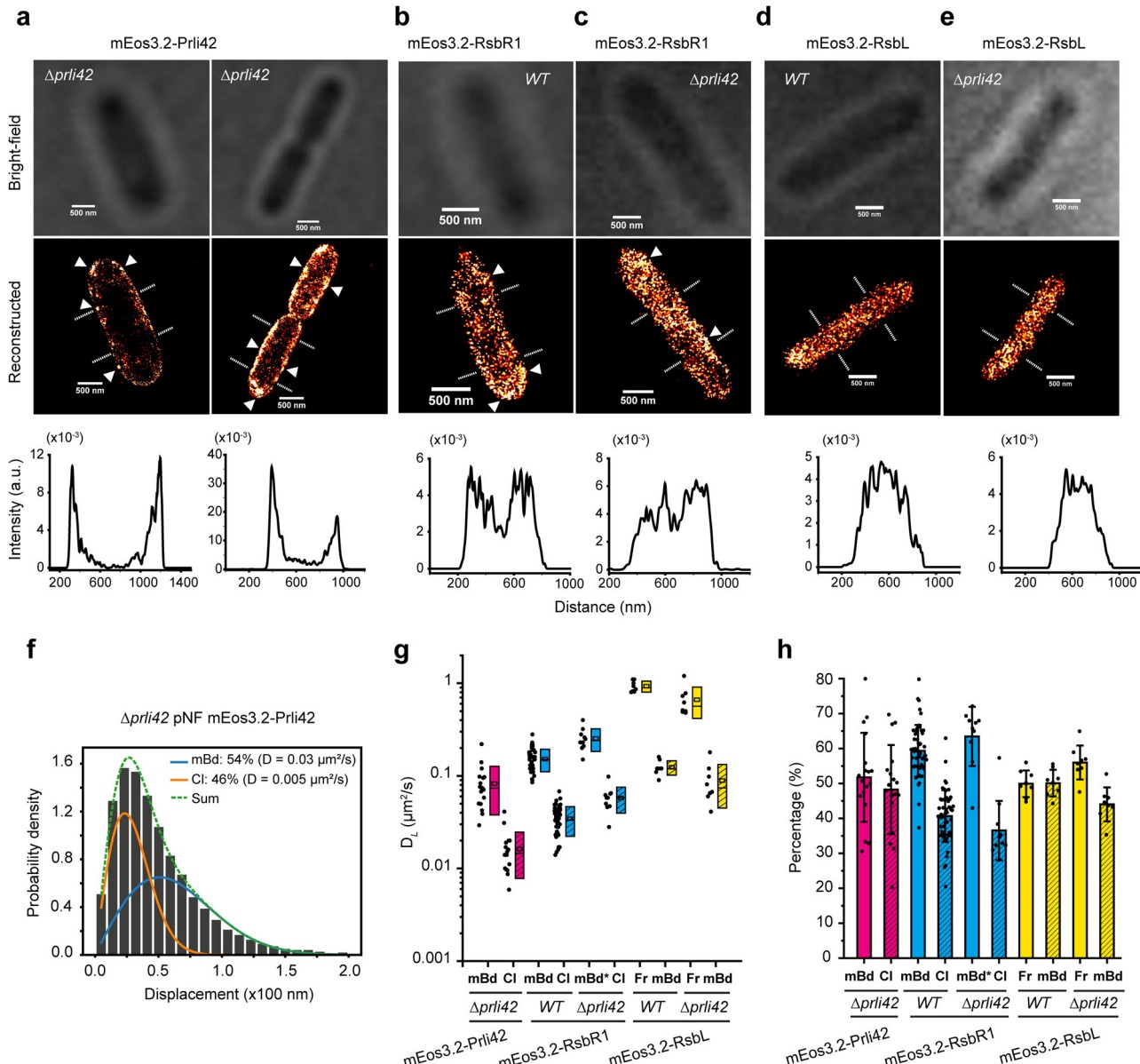

**Fig. 4 PALM localization and diffusion revealed by single-particle tracking. a**, Membrane localization of mEos3.2-Prli42 acquired from 2000 frames (left) and 5000 frames (right). **b**, and **c**, Localization of mEos3.2-RsbR1 in wild-type and Δ*prli42* strains, respectively (5000 frames). **d**, and **e**, Localization of mEos3.2-RsbL in wild-type and Δ*prli42* strains, respectively (5000 frames). The top and middle panels are bright-field and reconstructed images; Scale bars are 500 nm. Triangle symbols indicate high-density clusters. The cross-section profiles are shown in the lower panels. **f**, Distribution of single-molecule displacements of mEos3.2-Prli42. Blue and orange curves show the mBd and Cl fractions. The green dashed line is the sum of the two fractions. **g**, The $D_L$ of mEos3.2-Prli42 ($N = 18$), mEos3.2-RsbR1 ($N = 26$ and 10 for the wild-type and Δ*prli42*, respectively) and mEos3.2-RsbL ($N = 8$ for both wild-type and Δ*prli42*) were from cells in different fields of view (FOV) in three replicates. mEos3.2-Prli42 and mEos3.2-RsbR1 have membrane-bound (mBd) and clustered (Cl) fractions as shown on the x-axis; the starred membrane-bound fraction (mBd*) indicates mEos3.2-RsbR1 molecules loosely associated with membrane in the Δ*prli42* strain. mEos3.2-RsbL have free (Fr) and membrane-bound (mBd) fractions. Each dot shows the data of a FOV containing 10–50 well-separated cells. The box range indicates the standard deviation (SD), and the open square and dash symbols inside the boxes indicate the mean and median, respectively. **h** The corresponding percentage of the fraction in (**g**). Error bars represent standard deviations. Additional data on mEos3.2-RsbR1 and mEos3.2-RsbL localization and diffusion are in Supplementary Fig. 10.

acquisition time. The acquisition time is equal to the irradiation time. In the first 20 min (Fig. 6a), RsbL proteins are found evenly distributed in the cytoplasm and diffuse freely ($D_L = 1.84$ μm$^2$.s$^{-1}$) but considerably slower than free mEos3.2 ($D_L = 7.5 \pm 0.6$ μm$^2$.s$^{-1}$). There was a small (less than 16%) fraction of bound molecules in the first acquisitions, but after 40 min, this fraction increased to 38% with $D_L = 0.12$ μm$^2$.s$^{-1}$ (Fig. 6b). The slow diffusing component is associated with the periphery of the cells (hence, referred to as membrane-bound). In contrast, the molecules of the free fraction

(62%, $D_L = 2.03$ μm$^2$.s$^{-1}$) are primarily in the cytoplasm. Figures 6c, d are the SMdM results of the same cell of 6a and 6b after leaving in the dark for 70 min. The $D_L$ values and the fractions do not change significantly upon dark incubation, which is in line with data in Fig. 5b, c.

**Super-resolution images and SPT results of integrative strains.** We show PALM images of the integration strains *mEos3.2::rsbR1* and *mEos3.2::rsbL* in Fig. 7a, b. We applied continuous pre-irradiation

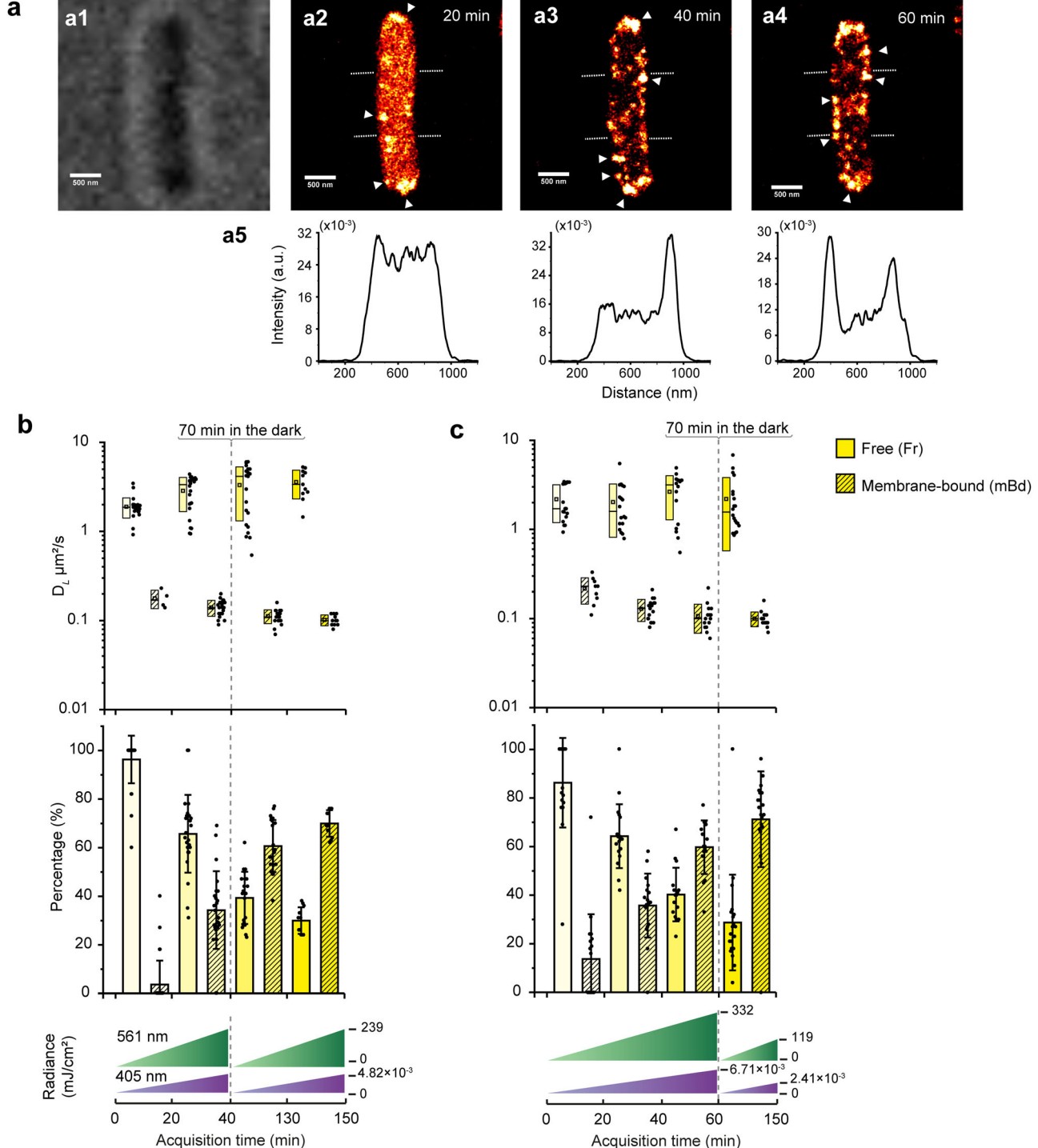

**Fig. 5 Effects of laser light irradiation during SMdM acquisition on localization and diffusion of mEos3.2-RsbL in *L. monocytogenes* EGD-e. a** Image of a single cell carrying pNF mEos3.2-RsbL in **(a1)** a bright-field. **(a2-4)** Reconstructed images of the same cell summing all single-molecule localizations of mEos3.2-RsbL of the wild-type strain after 20, 40 and 60 min of the irradiation during SMdM acquisition. (Scale bar 500 nm, Supplementary Video 3 for shorter time intervals). The $P_{dlt}$ promoter controls the mEos3.2-RsbL production on the plasmid. Triangle symbols indicate high-density clusters. The graphs below show the intensity profile across the cell **(a5)**. **b**, **c**, Clustering of RsbL upon irradiation (N = 28 and 18 after 20 min in **b** and **c**) and the test of dark recovery after 40 min (N = 24 and 18) and 60 min (N = 21 and 16) of SMdM acquisition, respectively. Top panel: box charts of the $D_L$ with the free (Fr) and membrane-bound (mBd) fraction. Each dot shows the data of one cell. The box range indicates the standard deviation (SD), and the open square and dash symbols inside the boxes indicate the mean and median, respectively. Middle panel: corresponding percentage of the fractions in the top panel. The sum of two fractions is 100%. Error bars represent standard deviations. The bottom panels are the radiance of 561 nm and 405 nm lasers measured at the focus above the glass slide during the acquisition. More details are in Supplementary Fig. 12. We paused the data acquisition after two measurements (40 min in **b**) (N = 11) or three measurements (60 min in **c**, N = 20), and left cells in the same field of view in the dark for 70 min. We attribute the clustering effect to irradiation.

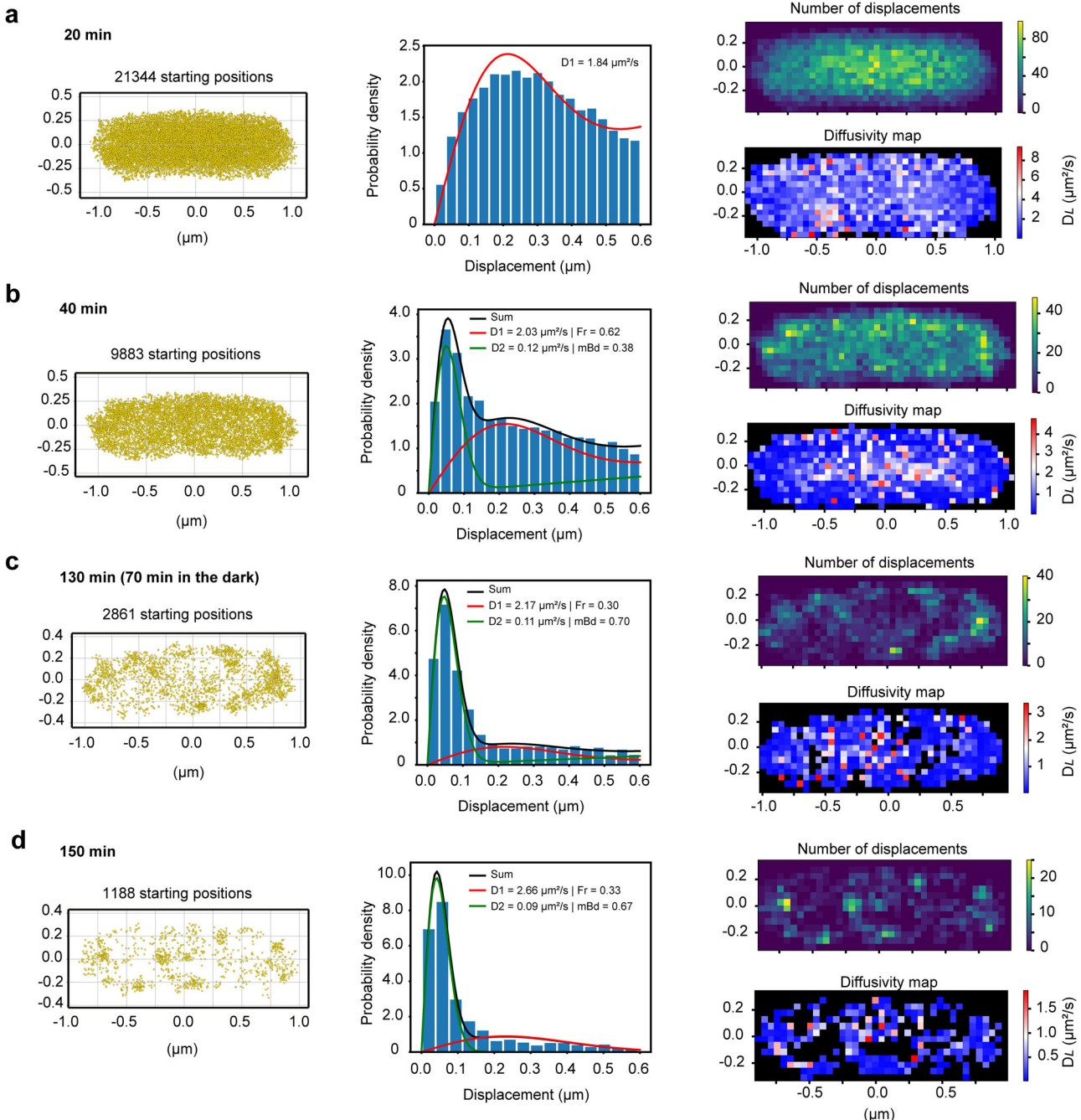

**Fig. 6 SMdM and 2-component fitting by maximum likelihood estimation (MLE). of mEos3.2-RsbL in *L. monocytogenes*. a–d,** SMdM results of a single cell over different data acquisition periods of 20 min, 40 min, 130 min (60 min plus 70 min in the dark), and 150 min, respectively. Left panel: a point cloud of starting localizations obtained by the Voronoï clustering method. Middle panel: distribution of all single-molecule displacements of the cell in the left panel. Red and green curves represent free (Fr) and membrane-bound (mBd) fractions with resulting $D_L$ and corresponding percentages, respectively, obtained from MLE for two components. Right (top) panel: spatial distribution of the number of displacements. Right (bottom) panel: Map of intracellular diffusivity with $50 \times 50 \ nm^2$ resolution.

of 405 nm for 10 s to photo-convert mEos3.2 as we anticipated a lower expression level from the native promoters on the chromosome. mEos3.2::RsbR1 was observed both in the cytoplasm and on the plasma membrane. We show three states with mEos3.2::RsbR1 not clustered, clustered or found mainly on the membrane. Also, for mEos3.2::RsbL, we detected clusters located on the plasma membrane. The results align with the data obtained from the plasmid-based expression of *mEos3.2::rsbR1* and *mEos3.2::rsbL*.

To determine the $D_L$ of proteins from the integrative strains, we used Eq. (6) with a 2-component model to fit the histogram of the molecule displacements. The distributions of single-molecule displacements from trajectories of mEos3.2::RsbR1 and mEos3.2::RsbL are shown in Fig. 7c, d, respectively. The $D_L$ and corresponding fractions of mEos3.2::RsbR1 and mEos3.2::RsbL obtained from SPT are shown in Fig. 7e, f. The $D_L$ values of mEos3.2::RsbR1 in the integrative strain are $0.12 \pm 0.05 \ \mu m^2.s^{-1}$ and $0.01 \pm 0.007 \ \mu m^2.s^{-1}$ for the membrane-bound (mBd) and clustered (Cl) molecules, respectively, each fraction is around 50%. The localization and diffusion results obtained with the integration strains of mEos3.2::RsbR1 agree with those from the

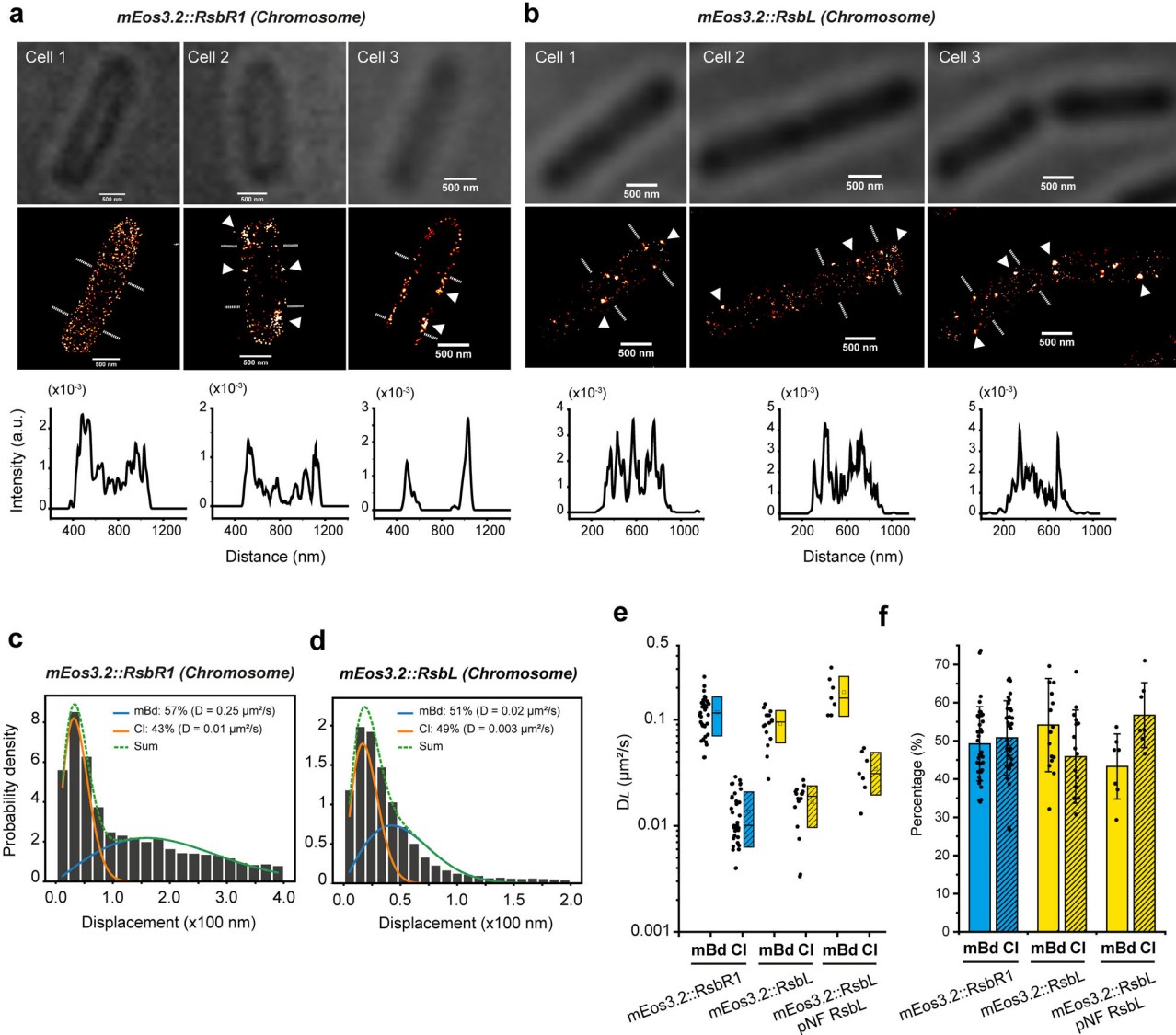

**Fig. 7 Protein localization by PALM and lateral diffusion by SPT in the integrative strains *mEos3.2::rsbR1* and *mEos3.2::rsbL*. a, b**, Localization of mEos3.2::RsbR1 and mEos3.2::RsbL, respectively. The top panels are bright-field images. Middle panels are reconstructed images from a series of frames. The native promoters on the chromosome control the expression of constructs (Scale bars 500 nm). Triangle symbols indicate high-density clusters. The graphs below show the cross-section profiles. **c**, and **d**, are the distribution of single-molecule displacements of mEos3.2::RsbR1 and mEos3.2::RsbL. Blue and orange curves, representing mBd and Cl, were obtained by fitting the histogram to the two-component 2D random walk model with Eq. (6). The green dashed line is the sum of two fractions. **e**, The $D_L$ of chromosome produced mEos3.2::RsbR1 ($N = 35$), mEos3.2::RsbL ($N = 16$), and mEos3.2::RsbL/pNF RsbL ($N = 7$) with membrane-bound (mBd) and clustered (Cl) molecules. Each dot shows the data of a FOV containing 10 – 50 well-separated cells. Box range indicates the standard deviation (SD), and open square and dash symbols inside the boxes indicate the mean and median, respectively, with the corresponding percentages shown in **f**. Error bars represent standard deviations.

vector-based expression. In the integrative strain of mEos3.2::RsbL, the $D_L$ values for mBd and Cl fractions are $0.10 \pm 0.03$ $\mu m^2.s^{-1}$ (54%) and $0.02 \pm 0.007$ $\mu m^2.s^{-1}$ (45%), respectively. Because of the abolished phenotype of the integrant *mEos3.2::rsbL*, the gene *rsbL* was added in trans on pNF vector, and in the complementary strain *mEos3.2::rsbL*/pNF *rsbL*, the $D_L$ were measured at $0.18 \pm 0.07$ $\mu m^2.s^{-1}$ (43%) and $0.03 \pm 0.01$ $\mu m^2.s^{-1}$ (57%) for mBd and Cl fractions, respectively. The clustering result of the integrative mEos3.2::RsbL upon irradiation follows the results obtained from the plasmid expression.

## Discussion

We have investigated the dynamics of proteins in *L. monocytogenes* focusing on the stressosome protein RsbR1 and the blue-light sensor RsbL and made the following important observations: (i) the small hydrophobic protein Prli42 plays a role in membrane localization of RsbR1 but does not influence stress sensing by RsbR1; (ii) RsbR1 has three diffusive states that may correspond to different biologically relevant states, including intermediate (protomers of RsbR1-RsbS), membrane-bound (via Prli42), and clustered (most likely in the form of stressosome complexes near the membrane) molecules; and (iii) the blue-light receptor RsbL diffuses in a free state but forms membrane-proximal clusters upon illumination, a process that is independent of Prli42.

The slower diffusion result of small molecules (i.e. mEos3.2) in the pole regions of *L. monocytogenes* is consistent with a recent SMdM study on the diffusion of proteins in the cytoplasm of *Escherichia coli*[42] and single-molecule tracking studies[37]. We have

not investigated the differences in pole and mid-cell diffusion in *L. monocytogenes* but the underlying mechanism(s) may be similar to that in *E. coli*. The slower diffusion is supposedly due to dynamic structures at the cell poles, such as protein aggregates (from ageing cells) or translating ribosomes and polysomes that hinder diffusion.

In the case of Prli42, the two components of diffusion in the membrane are consistent with a fraction of monomeric protein (or small oligomers) and clusters of Prli42 or Prli42-stressosome complexes. Various studies have shown that integral membrane proteins up to a size (radius, $R$ of 7 nm) diffuse according to the Saffman-Delbrück relationship[43,44], in which the $D_L$ scales with $\ln(1/R)$. Hence, the order of magnitude difference in $D_L$ implies substantial differences in the order of the oligomeric state of Prli42, and most likely, the slow diffusion reflects Prli42-stressosome complexes.

The association of RsbR1 with the plasma membrane is Prli42 dependent, but the differences in diffusion are small (Fig. 3f), suggesting that other, yet-to-be-identified, membrane proteins may also play a role in the localization of stressosome components. Moreover, a fraction of RsbR1 is found in the form of protomers and clusters in the cytoplasm. The finding that mEos3.2-RsbR1 localizes to the membrane shows that the mEos3.2 N-terminal tagging does not interfere with the Prli42 interaction. These results are in accordance with a recent cell fractionation study where 60% of RsbR1 membrane-associated[28]. Most RsbR1 was found on the membrane when *L. monocytogenes* resides in the intracellular eukaryotic environments[29]. Although Prli42 is involved in anchoring RsbR1 to the membrane, the physiological role of the membrane localization is not clear. We find that the growth curves of wild-type and Δ*prli42* cells are indistinguishable in different stress conditions (Supplementary Fig. 2). We also emphasise that Δ*prli42* is less susceptible to acidic shock at pH 2.5 than Δ*rsbR1* and Δ*sigB*. The Δ*prli42* mutant is not more sensitive than the wild-type, showing that it does not play a role in surviving acidic stress. (Supplementary Fig. 3). Thus, RsbR1 or SigB is more critical for stress sensing and response than the Prli42 membrane anchor. Since a fraction of RsbR1 is present in the cytoplasm irrespective of the presence of Prli42, it is possible that Prli42 serves as a hub to recruit excess RsbR1 and RsbR-like proteins and may not be involved in stressosome complex formation.

Cytosolic RsbR1 diffuses much slower than expected for free cytosolic protein with a molecular weight of 57.8 kDa (mEos3.2::RsbR1). The $D_L$ falls in the range of characterised DNA/RNA-binding proteins[38,45], ribosomes and ribosome-binding proteins[37,39,40,46], which suggests that RsbR1 is present in protomers and clusters. We also determined the mobility of RsbR1 under mild acid stress conditions (pH 5.0) as the protein is involved in acid stress sensing in *L. monocytogenes*[30]. The $D_L$ of cytoplasmic RsbR1 at pH 5.0 decreases compared to that at pH 7.2, which may reflect a different physical state (phase) of the cytoplasm at low pH[47–49]. This decrease in mobility is independent of Prli42, which argues against further association of RsbR1 with the membrane.

The sensing of blue light in *L. monocytogenes* is mediated by the blue-light sensor RsbL (Lmo0799). RsbL exhibits similar spectroscopic properties and blue-light-induced photochemistry with the reversible formation of a blue-shifted photoadduct as the ortholog YtvA in *Bacillus subtilis* and plant phototropins[35,50–52]. RsbL is also a positive effector of the general stress transcription factor sigma B in *L. monocytogenes*[53]. We now report that RsbL distributes uniformly in the cytoplasm and diffuses in the dark with a $D_L$ of $2.36 \pm 0.83\ \mu m^2.s^{-1}$. Upon irradiation with lasers (405 nm and 561 nm) during SMdM acquisition, mEos3.2-RsbL starts forming clusters that diffuse with an apparent $D_L$ of $0.14 \pm 0.04\ \mu m^2.s^{-1}$. The clustering behaviour is observed under conditions of overexpression and native expression of RsbL. It remains to be determined whether clustering is a property of unlabelled RsbL and

when it occurs whether it has a biological function. However, the light-dependent clustering was only observed for mEos3.2-RsbL and not for mEos3.2-RsbR1, suggesting that the RsbL undergoes conformational changes relevant to its functions.

The incorporation of blue-light sensors into the stressosome has previously been suggested by others[51,54,55]. Thus, RsbL may be present in *L. monocytogenes* in both monomeric (or lower-order oligomers) and higher-order aggregate states, depending on the illumination conditions[35]. The YtvA dimer from *Bacillus subtilis* forms heterotetrameric complexes with the RsbRA dimers, decreasing mobility upon blue light illumination[41]. For *L. monocytogenes* and based on proteomic and immunological assays, RsbL probably locates exclusively on the membrane after stressosome association[27,29]. We find that the strain lacking Prli42 (Δ*prli42*) also forms mEos3.2-RsbL clusters upon irradiation, and the two components of diffusion are comparable in wild-type and Δ*prli42*. Therefore, the clustering of RsbL upon irradiation is likely stressosome-related but not dependent on Prli42.

In the photochemical excitation of RsbL and YtvA, the ground state (dark state) absorption spectrum has features near 375, 450, and 475 nm[35,52]. After light excitation, RsbL and YtvA have an absorption peak at 380 nm. The photoconversion from the excited state to the ground state happens near UV light irradiation (356 and 405 nm)[56], i.e. conditions similar to the ones used by us to study the movement dynamics of RsbL.

We have observed clustering of mEos3.2-RsbL upon light excitation, but the SigB-dependent ring formation phenotype activated by RsbL was no longer seen. With the $D_L$ around $0.1\ \mu m^2.s^{-1}$ and $0.02\ \mu m^2.s^{-1}$ for membrane-bound and clustered fractions, mEos3.2-RsbL may associate with supramolecular complexes without light-activation of the stressosomes. This result suggests that light sensing by RsbL is a two-step process in which the clustering of RsbL is independent of the activation of the protein.

RsbL forms clusters upon illumination, which is entirely novel to the current knowledge. Unfortunately, based on the diffusion and localization data, we cannot conclude that RsbL is associated with the stressosome. However, since RsbL is one of the RsbR paralogs, it is likely that RsbL forms protomers with RsbS in a similar way to RsbR1-RsbS and forms stressosome complexes. Therefore, there are two mutually exclusive hypotheses for the clustering of RsbL upon illumination: (1) RsbL is associated with stressosome complexes or (2) the clustering upon illumination could be independent of stressosomes—a newly discovered property for RsbL.

In summary, we have uncovered the dynamics of stressosome proteins RsbR1 and RsbL under physiologically relevant conditions in the live cells of *L. monocytogenes*. On the basis of the diffusion components, we propose that RsbR1 clusters are in dynamic equilibrium with RsbR1-RsbS protomers and full stressosome complexes. Prli42 is involved in membrane localization of RsbR1, but we do not observe phenotypic differences upon deletion of the *prli42* gene. We also observe that the blue-light sensor RsbL, which in the dark diffuses freely in the cytoplasm, forms clusters upon irradiation. The present study lays the foundation for future research into the spatial and temporal distribution of stress-sensing components in living cells and the mechanism of signal transduction via the stressosome pathway in *L. monocytogenes*.

## Methods

**Bacterial strains, plasmids and primers, and growth conditions**. *L. monocytogenes* EGD-e and *Escherichia coli* strains, and plasmids and primers used in this study are listed in the Supplementary Table 2 and 3, respectively. We handled all cultures of *L. monocytogenes* EGD-e strains at biosafety level II (BSLII). In all experiments, we first streaked −80 °C glycerol stocks from a freezer to the Brain

Heart Infusion (BHI) agar plates to obtain a single colony. Cells from a single colony were inoculated in 3 mL BHI medium in 10 mL culture tubes, and then incubated overnight at 30 °C with 200 rpm shaking. The next day, the culture was diluted 1:100 in a fresh pre-warmed BHI medium and grown until $OD_{600}$ reached 0.6–0.8; unless indicated otherwise, we prepared the cells for microscopy measurements within this range of $OD_{600}$. The bacterial cells were collected by centrifugation at $7000 \times g$ for 2 min, then washed and suspended in PBS buffer (pH 7.4, 0.28 Osm) to obtain $OD_{600}$ ~2. Five microliter aliquots were taken for the measurement.

**Construction of genomic integrations in *L. monocytogenes* EGD-e.** We used the temperature-sensitive shuttle vector pMAD to construct knock-in strains expressing *rsbR* genes fused to the gene encoding the photo-convertible fluorescent protein mEos3.2[57]. The *mEos3.2* gene[58] was fused to *rsbR1* and *rsbL*, yielding tandem fusions of *mEos3.2::rsbR1* and *mEos3.2::rsbL* in pMAD, which were transformed into an *E. coli* K12 strain. In each fusion, six innocuous amino acids (GGTGGS) were inserted as a linker connecting the two proteins, with mEos3.2 at the N-terminal end. N-terminal fusions avoid interference with the binding of RsbS to the C-terminus of RsbR, which is required for stressosome formation[21]. The first three bases coding the methionine of RsbR1 and RsbL were removed to prevent alternative translation initiation. DNA sequencing confirmed the correctness of constructs, and the pMAD shuttle vectors were transformed into *L monocytogenes* EGD-e, which were allowed to integrate into the chromosome at the homologous locations and then followed with the excision of the vector. The expression of the fusion constructs is under the control of the native promoters. The transformation, integration, and excision were evaluated by consecutive screening methods, including blue-white screening, antibiotic sensitivity, and DNA sequencing. We followed the previously described protocols[59] to prepare competent cells and transform *L. monocytogenes* EGD-e.

**Construction of vectors with high expression.** We used the vector pNF8 with the constitutive $P_{dlt}$ promoter to obtain high protein production[60]. The *gfp* gene was replaced by the tandem fusion of *mEos3.2-rsbL* and *mEos3.2-rsbR1-rsbS-rsbT*, yielding vectors pNF mEos3.2-RsbL and pNF mEos3.2-RsbR1, respectively. The sequences of the fusions in the vectors were identical to those of the chromosomal integrations. For control experiments with free cytosolic mEos3.2 and membrane-bound Prli42 tagged with mEos3.2, we constructed pNF mEos3.2 and pNF mEos3.2-Prli42, respectively. All constructs were made by conventional restriction-cloning methods except for pNF mEos3.2, for which we used the USER® fusion method[61].

**In-gel fluorescence.** We performed in-gel fluorescence assays to examine the quality of proteins synthesized using a simple gel-based protocol as described previously[34], which discriminates folded from misfolded protein and breakdown products. In short, cultures in liquid BHI supplied with erythromycin (15 μg.mL$^{-1}$) were collected at $OD_{600} = 0.6$. Whole-cell samples were suspended and concentrated to $OD_{600} = 6$ in ice-cold 50 mM KPi (pH 7.2), 1 mM MgSO4, 10% (w.v$^{-1}$) glycerol, 1 mM PMSF plus trace amounts of DNase. The cell samples were disrupted by glass beads (300 mg, 0.1 mm diameter) in a cell disruptor (Disruptor Genie™) shaken for 5 min at 2840 rpm. After cooling the samples for 5 min on ice, the procedure was repeated one more time. We mixed 40 μL aliquots with 10 μL of 5× protein sample buffer containing 120 mM Tris-HCl (pH 6.8), 50% glycerol, 5% β-mercaptoethanol, 2% (w.v$^{-1}$) SDS, and 0.1% (w.v$^{-1}$) bromophenol blue. The mixtures were rested on ice until use. We analysed protein samples by 10% SDS/PAGE, and fluorescence of fusions tagged with mEos3.2 was excited by 488 nm wavelength (green state) and visualized with a LAS-3000 imaging system (Fujifilm).

**Phenotype test for integrative strain**
*Effect of different stresses on bacterial growth.* The test of bacterial growth upon exposure to environmental stress was done according as previously described[28]. Overnight cultures in BHI at 30 °C and 200 rpm were diluted 1:25 into BHI supplemented with ethanol (0% = control, 2%, 4%, and 8% v.v$^{-1}$); with NaCl (0% = control, 2%, 4%, and 8% w.v$^{-1}$); with $H_2O_2$ (0% = control, 0.05%, 0.1%, and 0.2% v.v$^{-1}$); and with HCl (for adjusting the pH to 7.2 = control, 5.5, 3.5 and 2.5). The cells were cultured in transparent 96-well plates and $OD_{600}$ values were measured using a TECAN microplate reader.

*Acid shock treatment.* We performed the acid shock treatment to test the acid tolerance response (ATR) phenotype of the integrative strain *mEos3.2::rsbR1* as described previously[30]. Briefly, overnight cultures grown at 30 °C were diluted in fresh BHI ($OD_{600} = 0.05$) and allowed to grow at 30 °C until the mid-log phase was reached ($OD_{600} = 0.4$). Cultures were separated into 2 tubes: one tube was acidified with HCl 5 M to obtain pH 5 to induce sigma B-dependent ATP[62], and one tube was kept at pH 7.2 (control). Both tubes were further incubated for 15 min at 30 °C. Cultures in both tubes were diluted 1:10 in BHI pH 2.5. Samples were taken at 0, 10, 20, and 30 min, serially diluted into PBS (pH 7.4, 0.28 Osm) and then plated on BHI agar. Plates were placed in a 30 °C incubator, and colonies were counted after 24 h. Three biological replicates were made.

*Ring phenotype of the blue-light sensor.* We tested the phenotype of the integrative strain *mEos3.2::rsbL*, using the ring formation assay upon light/dark cycles

previously described[31,53]. Overnight cultures were grown in BHI at 30 °C, standardized to $OD_{600} = 3$, and 2-μL aliquots were spotted on BHI plates containing 0.3% agar. Colonies in the semi-solid agar were allowed to grow at room temperature. The plates were either exposed under white fluorescent light with a power density of 100 to 200 μW.cm$^{-2}$, or were left in the dark (covered by aluminium foil) creating light/dark cycles with intervals of 12 h. Plates were photographed using the VWR® Gel Imager.

**Glass-slide treatment and cell immobilization for microscopy.** In PALM measurements, the immobilization of bacterial cells on a glass slide is crucial. To treat the glass slide, we applied (3-aminopropyl) triethoxysilane (APTES, Sigma-Aldrich) to treat the glass surface. High-precision glass slides (24 × 60 mm, 170 ± 5 μm thickness) were first sonicated in 5 M KOH for 1 h and rinsed thoroughly with Milli-Q. The slides were then dried using pressurized air. After drying, the surface was activated by oxygen plasma (65 W) for 1 min before being deposited in 1% APTES in Milli-Q for 20 min at room temperature. The glass slides were rinsed thoroughly using Milli-Q and finally dried by a pressurized air gun. To maintain appropriate moisture and keep cells in a physiologically relevant state during the microscopy measurements, we used an agarose pad (Duchefa Biochemie) (1% w.v$^{-1}$) in 2× PBS, which was placed on top of the bacteria cells and the glass slide (agarose pad-cells-glass slide)[59].

**FRAP acquisition and data analysis.** Using a previously described procedure, we used FRAP to measure the ensemble mobility of proteins in *L. monocytogenes*[63]. We determined the $D_L$ values of proteins in the exponential and stationary phase. The measurements were carried out on a Zeiss LSM710 confocal laser-scanning microscope (Zeiss, Oberkochen, Germany) with a C-apochromat 40× water immersion objective with NA of 1.2. We exploited the green state of mEos3.2, so the photobleaching (high intensity) and imaging (low intensity) were conducted at 488 nm. The fluorescent emission was collected from 493 to 700 nm. Half of a cell at one of the poles was designated as the bleaching area. The fluorescent profiles before and after the bleaching step were recorded (Supplementary Fig. 14). We used a home-written script in Python to analyse the FRAP data[63].

**Photoactivated localization microscopy (PALM).** We used a home-built inverted wide-field microscope Olympus IX-81 model equipped with a high numerical aperture objective (100×, NA = 1.49, oil immersion, Olympus) for super-resolution microscopy. Images of single molecules were captured using a 512 × 512 pixels EMCCD camera (C9100-13, Hamamatsu). The microscope and camera were turned on for three hours in advance to stabilize the ambient and stage temperature. In all measurements, the temperature of the stage was maintained at 21 ± 1 °C. We used solid-state lasers 405 nm (OBIS, 100 mW max. output) and 561 nm (Sapphire 561, 135 mW max. output) purchased from Coherent® (Santa Clara, USA). Laser beams were collimated with lenses and combined using dichroic mirrors. Laser powers were set at sources to 75 and 2 mW resulting in a power density on top of the glass surface of 3.56 mW.cm$^{-2}$ and $7.2 \times 10^{-5}$ mW.cm$^{-2}$ for 561 and 405 nm, respectively. A density filter was used for the laser 405 nm. We collected light emitted between 610 and 680 nm (ET 645/75 m filter, Chroma), and the illumination was done by highly inclined and laminated optical sheet (HILO) microscopy[64]. The HILO illumination with 2-dimensional projection and analysis of the membrane diffusion along the curved surface of the bacterial membrane makes that the $D_L$ values are likely underestimated by ~30% compared to the actual values in 3-dimension as suggested from previous studies[65,66].

**Single-molecule displacement mapping (SMdM)**
*Data acquisition.* We selected the field of view (FOV) at 250 × 250 pixels (25 × 25 μm$^2$), which resulted in the shortest possible exposure time of 17.08 ms and camera dead time of 0.78 ms, and, hence, the total frame time was 17.86 ms (~ 56 Hz). The autofocus function of the microscope was enabled to avoid z-drift. The modified stroboscopic laser illumination has been described in previous studies[32,42]. We set the pulse duration of the 405 nm and 561 nm lasers at 1 and 0.5 ms, respectively (Fig. 2a). The excitatory peak-to-peak time was tuned between 1.5 and 10 ms for (fast) cytosolic and (slow) membrane-bound proteins, respectively[32,67]. The frame time and pulse duration were verified by using an oscilloscope (Hameg Instruments, 100 Mhz analog scope, HM1004). The control of the camera and laser illuminations were synchronized by a custom Python script (https://github.com/MembraneEnzymology/smdm/tree/main/Microscopy). In the laser modulation script, we applied the *nidaqmx* package for interacting with the National Instruments current-generation data acquisition (NI-DAQmx) driver (https://github.com/ni/nidaqmx-python). The laser pulses were synchronized to the electrical pulses generated by the camera at the beginning of the dead time. A programmable card (National Instruments, PCI – 6602) was used to read the frame time and to modulate the laser pulses so that the frames were paired properly as shown in Fig. 2a. We acquired 2 consecutive movies per FOV for mEso3.2, mEo3.2::Prli42 and mEos3.2::RsbR1. Each movie had 65000 frames (~20 min). In case of mEos3.2::RsbL, we acquired 3-4 movies per FOV. The movies were saved as a MetaMorph® stack (.stk) (Molecular Devices), and concatenated in correct order as one movie, which was then converted to a tag-image file format (.tiff). We use

the background intensities to verify the correct order of frames after the concatenation, which was different for the odd and even frames.

*Single-molecule localization (peak detection).* For the compatibility between single-molecule detection and other steps in the SMdM analysis protocol, we employed the *STORM-analysis* package developed by the Zhuang lab (http://zhuang.harvard.edu/software.html) for peak detection. We used the *3D-DAOSTORM* program without incorporating the z-dimension since we are analysing in-plane 2D data[62,68]. Following the peak detection, the coordinates were corrected for xy-drift.

*Cell detection (point cloud clustering) and rotation.* The localized molecules were detected as coordinates and are displayed as points in Fig. 2b. The point clouds were clustered based on their densities using the Voronoï tessellation method[69] (Fig. 2c). We exploited the integrated Python library SciPy for the Voronoï clustering[70]. Each cluster indicates a cell, constituted by eigenvectors retrieved from the covariance matrix. The eigenvector information was used for the rotation and alignment of the clusters along the x-axis, which facilitates the pixelation and reconstruction of the diffusion map (Fig. 2d). In the diffusion map, we set the pixel size at $50 \times 50$ nm$^2$, and binned displacements in each pixel based on their starting positions. The $D_L$ values of the pixels were then obtained after fitting the binned displacements to the appropriate model (see below; Eqs. (1)–(4)).

*Peak pairing.* The displacement of the proteins in the 2D model was determined as a travel distance of a single-molecule in the fixed time interval between two excitatory peaks in a pair of frames. In each pair of frames, we matched the localized molecules in the first frame with the localized molecules in the second frame within a maximum radius of $r_{max} = 600$ nm. Cases with more than one localization within the searching radius create a pairing uncertainty. We corrected the uncertainty by including a background term in the fitting of the model (see Eq. (2)).

*Data fitting.* The analysis of the probability distribution is based on the single-molecule displacements as a function of fixed time interval ($\Delta t$). The calculation of the (quasi) diffusion coefficient from the displacement distribution of continuous long trajectories has been described previously[71–73]. We fitted displacement distributions to the probability density function $P(r,\Delta t)$ for a model of 2D random-walk diffusion:

$$P(r, \Delta t) = \frac{2r}{4D\Delta t} e^{-\frac{r^2}{4D\Delta t}} \tag{1}$$

Where $D$ is the diffusion coefficient; $r$ is the peak-to-peak displacement distance; and $\Delta t$ is the fixed time interval. $P(r,\Delta t)$ describes the Rayleigh distribution[74]. As for cases of pairing uncertainty, we followed a solution in which the background effect is incorporated. The solution presumes that within the searching radius the distribution of "background" molecules is homogenous and the probability of detecting a random "background" molecule linearly increases with $r$[32,67]. Thus, the background effect term $br$ was added to the Eq. (1):

$$P_1(r, \Delta t) = \frac{2r}{4D\Delta t} e^{-\frac{r^2}{4D\Delta t}} + br \tag{2}$$

Where $P_1(r,\Delta t)$ is the probability density with background correction; $b$ represents the slope of the background line. As we set the cut-off for searching radius at $r_{max} = 600$ nm, Eq. (2) was normalized to obtain the probability density $P_2(r,\Delta t)$ within the range $0 \le r \le r_{max}$, which is shown by:

$$P_2(r, \Delta t) = \frac{\frac{2r}{4D\Delta t} e^{-\frac{r^2}{4D\Delta t}} + br}{1 - e^{-\frac{r_{max}^2}{4D\Delta t}} + \frac{b}{2} r_{max}^2} \tag{3}$$

Equation (3) can be elaborated for data with multiple components as shown in Eq. (4):

$$P_3(r, \Delta t) = \frac{\sum_{i=1}^{N} f_i \frac{2r}{4D_i\Delta t} e^{-\frac{r^2}{4D_i\Delta t}} + br}{1 - \sum_{i=1}^{N} f_i e^{-\frac{r_{max}^2}{4D_i\Delta t}} + \frac{b}{2} r_{max}^2} \tag{4}$$

Where $P_3(r,\Delta t)$ is the global probability density for multiple components within the range $0 \le r \le r_{max}$; $N$ is the number of components; $f_i$ is the fraction ($\sum f_i = 1$); and $D_i$ is the corresponding diffusion coefficient. We used Eq. (4) to fit the histogram of displacements via maximum likelihood estimation (MLE), which yielded the diffusion coefficients for the proteins of interest from the SMdM measurements[32,67]. As we used the random diffusion model in two-dimension, the mean square displacement ($r^2$) can be shown by the equation: $r^2 = 4D\Delta t$[71]. Assuming all displacements are at $r_{max} = 600$ nm, we obtain $D = 60$ and $9 \mu m^2.s^{-1}$ for $\Delta t = 1.5$ and 10 ms, respectively, which are an order of magnitude greater than $D_L$ values obtained from FRAP measurements for free cytosolic and membrane proteins of *L. monocytogenes*[63] and FRAP data. Hence, the cut-off at $r_{max} = 600$ nm should be sufficient to cover all the displacements from the proteins analyzed.

*Selection of regions of interest in the cell.* We selected the regions of interest along the cell length, where 20% of the length was designated to each pole (counted from the leftmost and rightmost x coordinates after rotation), and the remaining 60%

was designated as the middle region. Data of all regions were fitted using Eq. (4). We used 20% of the total length to identify each pole because the average cell dimensions are 2.02 μm for the length and 0.32 μm for the radius, by calculating the ratio between radius and length, one obtains $0.32/2.02 = 0.16$. Hence, the radius represents 16% of the total length. Since the radius is the parameter that determines the size of the hemisphere constituting the pole, we rounded up and chose 20% of the length as an approximation of the pole area.

**Super-resolution image reconstruction.** For short movies (2000 – 10000 frames) of localization and single-particle tracking (SPT) experiments, we used a custom-written plugin for ImageJ called Single Molecule Biophysics (https://github.com/SingleMolecule) to detect and track single molecules, and reconstruct super-resolution images. We followed the protocol described previously[75–77]. We exploited a 2D Gaussian function closely related to the model for single-molecule detection used in SMdM analysis and fit the point-spread-function (PSF) forms of photons captured by the EMCCD camera, using the Levenberg–Marquardt algorithm[78,79].

$$f(x, y) = B + Ae^{-\left(\frac{(x-x_0)^2}{2\sigma_x^2} + \frac{(y-y_0)^2}{2\sigma_y^2}\right)} \tag{5}$$

Where $B$ is the background pixel intensity; $A$ is the amplitude; $x_0$ and $y_0$ correspond to the center positions; $\sigma_x$ and $\sigma_y$ are the $x$ and $y$ spread of the PSF. The localized molecules, resulting in the form of sub-pixel coordinates in all frames, were subsequently stacked in a reconstruction image.

**Single-particle tracking (SPT).** We used SPT with longer exposure times (30 and 50 ms) to track slow-moving molecules. The excitatory 561 nm pulses (1 ms) were placed at the beginning of frames. We matched the localized molecules in the consecutive frames within the maximum radius of 800 nm and then selected trajectories that lasted from 5 to 30 frames. We also fitted the displacement (step-size) distribution to the Eq. (1) to extract the diffusion coefficient, yet modified it for multiple components as shown in Eq. (6):

$$P_4(r, \Delta t) = \sum_{i=1}^{N} f_i \frac{2r}{4D_i\Delta t} e^{-\frac{r^2}{4D_i\Delta t}} \tag{6}$$

Where $r$ is the displacement distance; $\Delta t$ is the exposure time; and $f_i$ is the fraction ($\sum f_i = 1$); and $D_i$ is the corresponding diffusion coefficient. Since the acquisition for SPT was shorter and the exposure time longer compared to SMdM, and trajectories with pairing uncertainty issues were discarded, it was not necessary to incorporate the background effect term (see above).

**Statistics and reproducibility.** All experiments were completed in at least three replicates. In the experiments of the protein diffusion in particular regions in the cells (poles vs middle region), the diffusion of mEos3.2-RsbR1 in different pH conditions, and the diffusion coefficients determined by FRAP, we used one-way ANOVA to test the null hypothesis of the population means at $P < 0.05$ level and followed by Tukey's post hoc test for multiple comparisons at $P < 0.05$, $P < 0.01$, and $P < 0.001$ levels. $P$ values from Tukey's test were set at five digits after the decimal point.

**Reporting summary.** Further information on research design is available in the Nature Portfolio Reporting Summary linked to this article.

## Data availability

All data needed to evaluate the conclusions in the paper are present in the paper and/or the Supplementary Data 1.

## Code availability

Code developed for SMdM and FRAP analysis is available at Github (https://github.com/MembraneEnzymology).

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

## Acknowledgements

We thank Duarte N. Guerreiro for discussions. We thank Pascal Cossart for sharing the strain *L. monocytogenes* EGD-e *Δprli42* and the plasmid pAD-Prli42-FLAG. This project has received funding from the European Union's Horizon 2020 Research and Innovation Program under the Marie Skłodowska-Curie grant agreement no. 721456. The EU Marie-Curie ITN project SynCrop (project number 764591) and an ERC Advanced grant "ABCVolume" (grant number 670578).

## Author contributions

B.M.T and B.P. conceived and designed the study. B.P., A.I. C.O., T.A., J.J. supervised the study. B.M.T performed experiments and collected data. B.M.T performed data analysis with contributions from D.S.L, C.M.P, W.M.S, and L.M. B.M.T. and B.P. wrote the paper with input from W.M.S., L.M., A.I., C.O., T.A., and J.J. All authors contributed to the article and approved the submitted version.

## Competing interests

The authors declare no competing interests.
