## [Peer Review File · Communications Biology]

**Response to reviewers**

**Reviewer #1 (Remarks to the Author):**

The authors measured the diffusion of several proteins related to the stressosomes in
*Listeria monocytogenes*. Although the techniques employed are modern and
experiments are carefully done, the results do not form a cogent story that provides
new insights into the structure or function of stressosomes. Thus, this work may not
be suitable for Nature Communications.

Specifically, the authors used a fluorescent protein, mEos3.2, to label the different
proteins and measured diffusion. How are their diffusion measurements, which seem
to be based on the discussion of individual proteins, relate to the “RsbR-RsbS 60-
protomers truncated icosahedron” structure of the stressosome? Their PALM images
of mEos3.2::rsbR1 did not appear to be integrated into clusters that can be identified
as stressosomes.

>>> Reply: We thank the reviewer for the appreciation of our work, in particular the
technical aspects of the advanced microscopy and data analysis. We disagree with
the reviewer that we do not present a compelling story. We acknowledge that some
results are surprising and differ from the current thinking, e.g. the membrane-bound
state of RsbR1 is not required for sensing, and RsbL forms clusters upon
illumination. We feel that this emphasises the importance of our measurements,
which are complementary to classical molecular biology and microbiology
approaches used in the past.

We studied the dynamics of stressosome proteins in live cells, and we infer from the
diffusion data whether the proteins form higher-order assemblies such as
stressosomes or bind to the membrane. We see no other direct way of measuring
the dynamics of stressosome complexes; one always has to tag one of the subunits.
The RsbR-RsbS 60-protomer truncated icosahedron structure is composed of
multiple subunits from the structural study¹ that can also be present in the cell as
individual proteins or subcomplexes. It is the diversity of (sub-)structures that we
have probed with advanced light microscopy techniques.

Indeed, it is surprising that, unlike mEos3.2-RsbL clustered upon illumination,
mEos3.2-RsbR1 did not always integrate into one defined cluster size,

corresponding to the full stressosome complex, but that sub-complexes can be found
as well (Supplementary Video 1 and Supplementary Video 2). We have modified the
text to explain the approach and observations better.

Additional issues with the major conclusions:

1. A local heterogeneity of diffusion in the cytoplasm of *L. monocytogenes* with free
mEos3.2 proteins in the middle of the cell diffusing faster than in the pole regions.
This is not related to the stressosome topic of this work.

>>> Reply: We agree on the point of local heterogeneity, and we have no longer
accentuated the differences in the mobility of mEos3.2 in the pole regions and
middle of the cell as one of the main findings. However, the measurements on free
mEos3.2 are essential. It is a benchmark for the fusion constructs and comparing the
protein diffusions in the cytoplasm and at the membrane.

2. the small hydrophobic protein Prli42 is essential for membrane localization of
RsbR1 but does not influence stress sensing.

The results are not very convincing. For example, the D values were $0.2 \pm 0.04 \mu\text{m}^2/\text{s}$
for the wild type and $0.29 \pm 0.11 \mu\text{m}^2/\text{s}$ for the removal of prli42. Also, the pH=5
results of $0.14 \pm 0.02 \mu\text{m}^2/\text{s}$ and $0.15 \pm 0.03 \mu\text{m}^2/\text{s}$ were essentially the same between
the wildtype and prli42 removal. They also showed PALM images of RsbR1, but
even in the examples given, the differences in membrane localization are not very
clear between the wildtype and prli42 removal.

>>> Reply: We fully understand the concern of the reviewer. The differences in
diffusion coefficients are minor, yet they are significant - based on the confidence
level of the statistical test ($P < 0.001$). Notably, the SMdM and SPT data are internally
consistent (Fig. 3f and Fig. 4g). The analysis of the protein diffusion that can be
present in multiple states is complicated², and that is why we used different
microscopy techniques (SMdM, SPT, FRAP and PALM) to substantiate our findings.
We have modified the manuscript (Lines 227 – 234 and 255 - 256) by emphasizing
the key findings, the significance of the data, and the intricacies of obtaining diffusion
coefficients of proteins that can be present in different aggregation states.
Additionally, Supplementary Videos 1 and 2 are added to show the localization and
clustering of RsbR1.

3. The cytoplasmic fraction of RsbR1 is present in supramolecular complexes,
presumably with RsbS (and probably RsbT) to form the stressosome;

These seem not directly supported by their diffusion data.

>>> Reply: We postulate the states of RsbR1 on the basis of its much slower
diffusion than expected for a free cytosolic protein with $M_w = 57.8$ kDa (mEos3.2-
RsbR1). We emphasise that the mEos3.2-RsbR1 fusion is in-frame with RsbS and
RsbT both on the chromosome and plasmid. Hence, RsbR1 is likely present in the
form of protomers of RsbR1-RsbS (but not in a full stressosome complex), which
explains the slow diffusion. We have changed and assigned this state as the
Intermediate state (Int), while monomeric mEos3.2-RsbR1 is the free state. We
rephrased the text to clarify our interpretation of the data (Lines 280 - 290, 304 -
309, 451 – 454, and 468 – 470).

4. The slowing of diffusion of the blue light receptor RsbL upon illumination suggests
that the association of the protein with the stressosome is blue-light dependent, a
process that is independent of Prli42.

They used 405 and 560 nm. Which of the two illuminations played the role? Neither
is “blue light”. Also, they should compare with another target like RsbR1 as a
negative control to show no similar slowing of diffusion after illumination.

>>> Reply: The name blue-light receptor is given for RsbL (Lmo0799) in *L.*
*monocytogenes* (and its homolog YtvA in *B. subtilis*) as it possesses a light, oxygen,
voltage (LOV) domain that is homologous to the photoactive, flavin mononucleotide
(FMN)-binding LOV domains of phototropin found in higher plants³. The
photochemistry of RsbL shows an absorption spectrum in the ground state (dark
state) with maxima near 375, 450, and 475 nm^{3,4}. We used the laser lines of our
microscopes to excite RsbL, which indeed does not correspond to the absorption
maxima but suffice for photoconversion between the different states. We thus
conclude that the clustering of RsbL, elicited by 405 nm illumination, is
physiologically relevant.

As for the controls - we show in the manuscript (Lines 317 - 319 and 497 - 517) that
the clustering upon laser illumination does not occur with free mEos3.2, mEos3.2-
Prli42, and mEos3.2-RsbR1 (Supplementary Videos 1, 2, versus 3). Hence, the
clustering of mEos3.2-RsbL is a genuine property of the light sensor RsbL.

**Reviewer #2 (Remarks to the Author):**

The authors of “The dynamics and function of stressosome proteins in *Listeria*
*monocytogenes*” present thorough subcellular analysis of the location and diffusion
of stressosome protein RsbR1 using a fluorescently tagged version. The techniques
used allow for detailed and sensitive measurements that have not been shown before
for the stressosome. Given the ubiquity of the stressosome in bacterial species and
its “stress sensing hub” role, questions about the cellular and subcellular behavior of
proteins in this complex are important ones to address. Overall the data of the
mEose3.2::RsbR1 construct provide interesting observations about the subcellular
localization and dynamics of RsbR1 (Fig3e, Fig3f, Fig4b, Fig4c) and the role that the
membrane bound protein Prli42 plays in the diffusion of RsbR1. The results are
consistent with those found previously in that RsbR1 interacts with Prli42 and the
membrane association of RsbR1 is dependent on Prli42. Unfortunately, this reviewer
found the data on RsbL less convincing and has suggestions on how to address it.
Overall, the RsbR1 dynamic characterization will be useful for further testing a
variety of SigB inducing conditions to understand stressosome dynamics.

>>> Reply: We thank the reviewer for his/her favourable evaluation of the paper and
valid suggestions.

Recommendations.

1. Move the construct strain characterization text to the beginning of the results
section. Lines 402-423. It is important to establish early on that the mEos3.2 fusions
are functional so that the localization data are interpretable. The authors show that
the chromosomal integrated RsbR1 construct phenocopies the WT in Supplementary
figure 12. But the data on the RsbL construct are less favourable showing the
construct is non-functional in its light sensing function, Supplementary Figure 13.
Given that the characterization of the RsbL fusion is done using light irradiation, and
cells containing the construct did not complement a null in a light sensing assay, the
data on RsbL localization and clustering are less than convincing. This reviewer
would recommend, removing the RsbL data all together or at the very least noting
that the data are inconclusive and deemphasizing them. See #2 for more details on
specific text language.

>>> Reply: We agree, and we have moved the phenotypic screening to the
beginning of the results section. We also agree to de-emphasize the data of RsbL
association with stressosome complexes. The appropriate changes have been made
at lines 86 – 88, 432 – 434, and 510 - 517.

2. This reviewer suggests that the authors remove language from the manuscript
that alludes to conclusions not shown by the data presented regarding RsbL
clustering. For example, lines 94, 95, 544, 569, 570. In these instances, the authors
state the clustering of RsbL is stressosome associated, yet no evidence is shown for
that conclusion. Data showing RsbL fusion clustering in a stressosome deficient cell
or other co-localization assays would give weight to those statements. Additionally,
the fact that the fusion is nonfunctional for light sensing makes the clustering
phenotype observed puzzling. Any affirmative conclusions about the RsbL fusion
need further evidence and are not supported as they currently stand in the
manuscript.

>>> Reply: We agree, and we have rephrased the text accordingly (Line 24 – 25, 86
143 – 88, 432 - 434, 522 - 524). The assumption that the clustering of RsbL might be
stressosome-associated is based on the change in the diffusion coefficient. We
acknowledge that it is unfortunate that RsbL fusions are no longer functional in light
sensing, yet the clustering is still light-dependent.

3. Given that the localization (membrane bound vs free) and diffusion rates of RsbR1
change in the *pri42* null strain, a complementary experiment would be to make the
N-terminal mutant of *pri42* known to prevent *Pri42* interaction with RsbR1 and
measure RsbR1 construct dynamics. This could refine the molecular model of
interactions between the two proteins or if found differently propose new hypotheses.

>>> Reply: The suggested experiment is important to understand the interaction
between *Pri42* and RsbR1 and has been done by immunoprecipitation and
immunoblotting in a previous study⁵. However, we found that the growth rates and
acidic stress responses of the wild-type and *pri42* null strain are identical in the
phenotypic screenings (See Phenotypic screening of the integrative strain,
Supplementary Fig. 11, and Supplementary Fig. 12). Hence, *Pri42* is important for
tethering the stressosome to the membrane but does not play essential roles in
sensing stress. The precise mode of action of *Pri42* remains to be uncovered. *Pri42*

has been hypothesized that it could bring the stressosome into contact with a
membrane protein, but that protein is as-yet-unidentified^{5,6}. Therefore, the work on
the complementary experiment is beyond the scope of our study.

4. One important question about stressosome structure is whether in vivo
stressosomes are heterogenous complexes and whether the stoichiometry of the
different RsbR paralogs changes during stress sensing and signaling. It seems that
the RsbR1 fusion characterization was done in a wild type strain (unless I missed it)
carrying the other RsbR paralogs, and therefore the RsbR1 localization involved a
stressosome containing other RsbR members. Dessaux et al showed in 2020 that
RsbR1 interactions are affected by the presence of its paralogs, what would happen
to RsbR1 dynamics when it is the only the sensor in the stressosome? This
experiment would add mechanistic insight to the phenomena observed by the
authors and strengthen the impact the results have.

>>> Reply: We appreciate the suggestion of the reviewer, but we also feel that it is
beyond the scope of this study.

**Reviewer #3 (Remarks to the Author):**

This manuscript by Tran et al. is both interesting and important, as it is at a key
frontier for studies of signal sensation and transduction by bacterial stressosomes—
namely, to visualize in vivo the molecular-level events that accompany stress
sensation and signaling. This report takes a step towards the goal of understanding
the dynamics of constituent proteins at the individual stressosome level to initiate the
SigB-mediated stress response. In this work, the authors use single-particle tracking
and other super-resolution fluorescence microscopic techniques to probe the
localization and motion of two stressosome-associated sensors (RsbR1 and the light
sensor RsbL) and one membrane-bound accessory factor (Prli42) that putatively
tethers stressosome-associated factors to the cell membrane. The experiments are
well done, and I am enthusiastic about the data.

Perhaps my biggest difficulty with this manuscript was that it does not follow a logical
progression, making it very difficult to understand. What is the central question of the
paper? There are many disparate results: from diffusion in different cell zones to
localization of different proteins, many different diffusion numbers, plasmid-borne

and integrated genes, but what is the overall message? I understand that some of
the findings are disparate, but for the sake of the reader the text needs to lead the
reader logically from question to experiment to answer.

In my reading, the most important and exciting findings of the study are 1)
confirmation that RsbR1 membrane-proximal location largely depends on Prli42; 2)
that mEos-RsbR1 shows two different diffusion coefficients, suggestive of different
association states; and 3) that mEos-RsbL diffuses faster than RsbR1 but clusters as
cells are irradiated with light, slowing RsbL diffusion to values similar to that of
RsbR1.

>>> Reply: We thank the reviewer for his/her favourable evaluation of the paper and
valid suggestions. A number of his/her comments have been addressed in the
rebuttal to R1 and R2.

Main concerns

1. It was unclear how some of the categories used in the paper were derived. Are the
“membrane-bound” and “clustered” fractions of the fluorescent proteins defined by
their diffusion coefficients or by their cellular localization? What were the criteria for
placing a particular particle into each category?

>>> Reply: Very good point, thank you. We added a paragraph to explain the
categories (Lines 280 - 290). SMdM and SPT provide both localization and mobility
data. We categorised the protein states on the basis of individual trajectories of
particles and the obtained diffusion coefficients; the latter gives an estimate of the
size. However, it is practically impossible to determine the exact number of diffusive
states on the basis of the tracking data². Hence, we first used the localization data
(membrane or cytoplasm; different conditions) and then the tracking (SPT) and
SMdM data to estimate the diffusion coefficients, from which we infer whether or not
the proteins cluster.

2. Because of its potential interaction with stressosomes, mEos-Prli42 diffusion is
likely not representative of a generic membrane protein, and its diffusion is on the
same order as the “mBd” and “Cl” fractions of R1 and L, consistent with an
interaction. To test how Prli42-stressosome interactions impact the diffusion of each,
a useful control would be a mutant of mEos-Prli42 that does not interact with RsbR1
(e.g., the R8A mutation; Impens et al. 2017).

>>> Reply: Membrane diffusion is largely determined by the viscosity of the
membrane (see also Table 1 in Doeven et al. 2005)⁷, and the diffusion coefficient of
membrane-bound proteins has a different dependence on size than of water-soluble
proteins (see Ramadurai et al (2009) JACS 131: 12650)⁸. The hydrophobicity
properties of Prli42 make it a genuine membrane protein, and the diffusion
coefficient of mEos3.2-Prli42 is in accordance with observations made for other
membrane proteins⁹⁻¹⁴. Therefore, we feel that making the R8A mutation would not
yield much additional information. (See comment 3 of Reviewer 2)

3. The finding that mEos-RsbR1 has two distinct diffusion coefficients that are
several-fold different is quite intriguing, because it suggests two different association
states. But the faster-diffusing population is still much slower (0.15-0.29 $\mu\text{m}^2/\text{s}$) than
free mEos and is not greatly impacted by the presence of Prli42, suggesting that
neither association state is free—are we seeing single stressosomes and then
stressosome clusters? How do the authors interpret this central finding of the study?

>>> Reply: We speculated that RsbR1 in the free (Fr) state is in the form of
protomers of RsbR1-RsbS with $D = 0.46 - 0.75 \mu\text{m}^2/\text{s}$, whereas Rsb1 as part of the
full stressosome complex (RsbR1-RsbS-RsbT) has $D = 0.01 - 0.03 \mu\text{m}^2/\text{s}$. We thank
the reviewer for pointing out that the “free state” is confusing in the context of
mEos3.2-RsbR1. We now refer to the component with $D = 0.46 - 0.75 \mu\text{m}^2/\text{s}$ as the
Intermediate state (Int), whereas monomeric mEos3.2-RsbR1 is the free state. We
have modified the text and figures accordingly.

We emphasise that when mEos3.2-RsbR1 is highly expressed from a plasmid
(without in-frame fusion to the genes for RsbS and RsbT) the molecules end up in
large immobile aggregates. This is the reason why subsequently we made
chromosomal integration of *mEos3.2::rsbR1* in-frame with *rsbS* and *rsbT*. For
mEos3.2-RsbR1 molecules at membrane-proximal locations (mBd fraction), we
observe $D = 0.12 - 0.29 \mu\text{m}^2/\text{s}$, which is similar to the diffusion of Prli42 and
suggests a fraction of monomeric mEos3.2-RsbR1 bound to Prli42 or unknown
membrane components. We provide additional Supplementary Videos 1 and 2 to
show the localization and clustering of RsbR1.

4. I am sympathetic to the intriguing idea that RsbL clustering might represent light-
dependent association of L with stressosomes. But two difficulties with this

interpretation are 1) that the “Cl” fraction of RsbL diffuses faster than the “Cl” fraction
of RsbR1 and 2) that R1 clustering is much rarer in the reconstructed images than L
clustering after irradiation. If the clustering of L is into stressosomes, why don’t we
see the same clustered pattern for R1? As this is another main finding of the paper, it
deserves careful interpretation. It is certainly possible that the clustering is
independent of stressosomes and is just a newly discovered property of RsbL after
light exposure—perhaps even a mechanism by which the light response is turned
off?

>>> Reply: The reviewer has partly given the answer him/herself. We added a
paragraph (Lines 510 – 517) to discuss the clustering effect of RsbL. Indeed, RsbL
forms clusters upon illumination, which is completely novel to the current knowledge.
Unfortunately, we cannot conclude whether RsbL is associated with stressosome
with the diffusion and localization data. However, since RsbL is one of the RsbR
paralogs, it is likely that RsbL forms protomers with RsbS in a similar way to RsbR1-
RsbS and form stressosome complexes. Therefore, there are two mutually exclusive
hypotheses for the clustering of RsbL upon illumination: (1) RsbL is associated with
stressosome complexes or (2) the clustering upon illumination could be independent
of stressosomes - a newly discovered property for RsbL.

Minor points

1. The title is vague—it sounds like a review. It should instead reflect the central
message of the paper (that’s up to the authors but should be specific). Two
examples: “RsbR1 of *Listeria monocytogenes* displays two diffusion states and is
membrane-localized by Prli42” or “Stressosome sensor proteins in *Listeria*
*monocytogenes* display clustering and membrane localization”. Or even
“Stressosome sensor proteins in *Listeria monocytogenes* show multiple diffusion
states in vivo”.

>>> Reply: We have changed the title to “Super-resolving microscopy reveals the
localizations and movement dynamics of stressosome proteins in
*Listeria monocytogenes*”.

2. The abstract and text sometimes make claims that are not strictly supported by
the data, e.g., l. 25-26 and 94-95. Association of RsbL with the stressosome complex

upon exposure to light is not shown here. The diffusion rates of clustered L are
perhaps consistent with stressosome interaction, but there are other possible
interpretations. Another small example is I. 422-423, that tagging of RsbL hampers
its light sensing. It seems clear that it hampers signaling to sigB, but the light-
dependent clustering result suggests that it can still indeed respond to (and thus
sense) light.

>>> Reply: We rephrased the text to make the points clearer.

3. Proper genetic nomenclature should be followed in the text and strain table (“:”
means an insertion).

>>> Reply: This is a valid point. We corrected the genetic nomenclatures in the text,
images, and tables.

References

- 1. Williams, A. H. *et al.* The cryo-electron microscopy supramolecular structure of
the bacterial stressosome unveils its mechanism of activation. *Nat. Commun.*
**10**, (2019).
- 2. Persson, F., Lindén, M., Unoson, C. & Elf, J. Extracting intracellular diffusive
states and transition rates from single-molecule tracking data. *Nat. Methods*
**10**, 265–269 (2013).
- 3. Losi, A., Polverini, E., Quest, B. & Gärtner, W. First evidence for phototropin-
related blue-light receptors in prokaryotes. *Biophys. J.* **82**, 2627–2634 (2002).
- 4. Chan, R. H., Lewis, J. W. & Bogomolni, R. A. Photocycle of the LOV-STAS
protein from the pathogen *Listeria monocytogenes*. *Photochem. Photobiol.* **89**,
361–369 (2013).
- 5. Impens, F. *et al.* N-terminomics identifies Prli42 as a membrane miniprotein
conserved in Firmicutes and critical for stressosome activation in *Listeria*
*monocytogenes*. *Nat. Microbiol.* **2**, (2017).
- 6. Radoshevich, L. & Cossart, P. *Listeria monocytogenes*: towards a complete
picture of its physiology and pathogenesis. *Nat. Rev. Microbiol.* **16**, 32–46
(2018).
- 7. Doeven, M. K. *et al.* Distribution, lateral mobility and function of membrane

- proteins incorporated into giant unilamellar vesicles. *Biophys. J.* **88**, 1134–
1142 (2005).
- 8. Ramadurai, S. *et al.* Lateral diffusion of membrane proteins. *J. Am. Chem.*
*Soc.* **131**, 12650–12656 (2009).
- 9. Oswald, F., Varadarajan, A., Lill, H., Peterman, E. J. G. & Bollen, Y. J. M.
MreB-dependent organization of the E. coli cytoplasmic membrane controls
membrane protein diffusion. *Biophys. J.* **110**, 1139–1149 (2016).
- 10. Kumar, M., Mommer, M. S. & Sourjik, V. Mobility of cytoplasmic, membrane,
and DNA-binding proteins in Escherichia coli. *Biophys. J.* **98**, 552–559 (2010).
- 11. Mullineaux, C. W., Nenninger, A., Ray, N. & Robinson, C. Diffusion of green
fluorescent protein in three cell environments in Escherichia coli. *J. Bacteriol.*
**188**, 3442–3448 (2006).
- 12. Seinen, A. B., Spakman, D., van Oijen, A. M. & Driessen, A. J. M. Cellular
dynamics of the SecA ATPase at the single molecule level. *Sci. Rep.* **11**, 1–16
(2021).
- 13. Van Den Berg, J., Galbiati, H., Rasmussen, A., Miller, S. & Poolman, B. On the
mobility, membrane location and functionality of mechanosensitive channels in
Escherichia coli. *Sci. Rep.* **6**, 1–11 (2016).
- 14. Mika, J. T., Schavemaker, P. E., Krasnikov, V. & Poolman, B. Impact of
osmotic stress on protein diffusion in Lactococcus lactis. *Mol. Microbiol.* **94**,
857–870 (2014).

REVIEWERS' COMMENTS:

Reviewer #1 (Remarks to the Author):

The authors have addressed my questions and the manuscript was improved. I recommend publication.

Reviewer #2 (Remarks to the Author):

In this revised manuscript by Tran et al., the authors have done a good job addressing many of the reviewer comments. The data that this paper contributes to the field are valuable, despite the fact that not all of the observed phenomena are yet fully explained or connected to other work on the stressosome. In the revision, the interpretations match the data rather than drawing inferences that are not formally supported by the results. The authors should be applauded for undertaking this study, which represents a very substantial amount of careful microscopic analysis, not to mention the genetic work. It will be interesting to gain in future studies a more complete cell biological picture of stressosome dynamics within cells. In my opinion, the paper is essentially publication ready. I have only a few minor textual recommendations.

Minor points

ll. 292, 440: Agreed that RsbR1 has three distinguishable diffusive states, but whether each diffusive state represents a biologically relevant (i.e., functionally distinct), state formally remains unknown, as there are no functional tests for each state nor phenotypic differences that are conditioned on a particular diffusive state. Thus I advocate that these statements be qualified (“...diffusive states that may correspond to biologically relevant differences...”, for example).

ll. 495-6, 506, 516-517, 524. In the interpretations regarding the interesting light-stimulated clustering of mEos3.2-rsbl, caution should be used to avoid drawing conclusions about the native function of RsbL (e.g., that light-sensing is a 2-step process that includes clustering). The data indicate only that mEos3.2-RsbL, which is nonfunctional based on the ring-formation phenotype, clusters. Hence it remains unknown whether clustering is a property of unlabeled RsbL and whether, if it occurs, whether it has a biological function or is incidental.